# Transition from positive to negative indirect $CO_2$ effects on the vegetation carbon uptake

Zefeng Chen [1,2,3], Weiguang Wang [1,2,3] ✉, Giovanni Forzieri[4] & Alessandro Cescatti[5]

Although elevated atmospheric $CO_2$ concentration (e$CO_2$) has substantial indirect effects on vegetation carbon uptake via associated climate change, their dynamics remain unclear. Here we investigate how the impacts of e$CO_2$-driven climate change on growing-season gross primary production have changed globally during 1982–2014, using satellite observations and Earth system models, and evaluate their evolution until the year 2100. We show that the initial positive effect of e$CO_2$-induced climate change on vegetation carbon uptake has declined recently, shifting to negative in the early 21st century. Such emerging pattern appears prominent in high latitudes and occurs in combination with a decrease of direct $CO_2$ physiological effect, ultimately resulting in a sharp reduction of the current growth benefits induced by climate warming and $CO_2$ fertilization. Such weakening of the indirect $CO_2$ effect can be partially attributed to the widespread land drying, and it is expected to be further exacerbated under global warming.

Terrestrial ecosystems absorb ~30% of anthropogenic carbon dioxide ($CO_2$) emissions and thus play a fundamental role in mitigating climate change[1,2]. Over the past five decades, the terrestrial carbon sink has more than doubled at a pace that is consistent with the increase in anthropogenic $CO_2$ emissions[2,3]. Current evidence demonstrates that the enhancement of the terrestrial carbon sink is partially attributable to the increased carbon uptake by vegetation under elevated atmospheric $CO_2$ concentration (e$CO_2$)[4–7]. The e$CO_2$-induced changes in vegetation carbon uptake (represented by gross primary production (GPP)) are governed by two different mechanisms. The first is the direct effect of e$CO_2$ through the stimulation of photosynthetic carbon fixation and the enhancement of water-use efficiency (hereafter e$CO_2$(dir))[8,9]. The second is the indirect effect of e$CO_2$ through its radiative forcing and the associated change in climate (e.g., temperature and water regime) and related environmental conditions (e.g., variation in nitrogen availability linked to temperature-driven changes in the mineralization rate of soil organic matter) (hereafter e$CO_2$(ind))[10–12]. Recently, data-driven assessments based on in-situ and satellite observations have documented a declining trend in e$CO_2$(dir)[9,13]. Given the dominant role of e$CO_2$ in the recent increase in

GPP[14], the sign and temporal variation in e$CO_2$(ind) is expected to increasingly control the future trajectory of the terrestrial carbon budget[15]. However, the dynamics of such indirect $CO_2$ effect on the terrestrial carbon budget remain largely elusive. The relative importance of future indirect versus direct effects of e$CO_2$ in regulating vegetation carbon uptake has not yet been quantified, and the underlying ecological mechanisms remain poorly understood. Such knowledge gaps are reflected in substantial uncertainties in the effectiveness of land-based climate mitigation policies.

e$CO_2$(ind) originates from the strong and non-linear effects of e$CO_2$-induced climate change on terrestrial GPP, which involve multiple pathways, including the plants' response to changing temperature, water supply, atmospheric dryness (expressed by vapor pressure deficit, VPD) and their complex interactions[16]. In addition, these pathways via which climate influences GPP also interact with the e$CO_2$(dir)[17,18]. For example, the rising VPD with e$CO_2$ generally causes a reduction in stomatal aperture, modulating the transpiration rate and —at the same time— the positive effect of $CO_2$ fertilization on photosynthesis[19]. In view of the variety of interacting feedbacks that regulate vegetation carbon uptake, it is challenging to quantitatively

[1]National Key Laboratory of Water Disaster Prevention, Hohai University, Nanjing, China. [2]Yangtze Institute for Conservation and Development, Hohai University, Nanjing, China. [3]College of Hydrology and Water Resources, Hohai University, Nanjing, China. [4]Department of Civil and Environmental Engineering, University of Florence, Florence, Italy. [5]European Commission, Joint Research Centre, Ispra, Italy. ✉e-mail: wangweiguang@hhu.edu.cn

disentangle the total $eCO_2(ind)$, particularly at regional-to-global scales, where local-scale findings of Free-air $CO_2$ enrichment (FACE) experiments may not be applicable[20,21]. Recent studies based on satellite products and model simulations have reported the weakening of the temperature-vegetation relationship in northern ecosystems over the past 30 years[22], the increasingly negative impact of VPD on alpine grassland productivity[23], and the increasing water constraint on vegetation growth in many regions across the globe[24,25] and the corresponding higher risk of droughts to the global carbon cycle[26]. However, considering that changes in temperature, atmospheric dryness, precipitation, and soil moisture are single components of the climate response to the $CO_2$ radiative forcing, findings of the above-mentioned studies can only partially reflect the temporal variations in $eCO_2(ind)$. $eCO_2$ drives changes in various climatic factors, and their effects on vegetation carbon uptake are covariant, can be additive or offsetting, and may lead to nonlinearities due to different feedback mechanisms[27,28]. Existing studies focusing on a single climate driver (e.g., temperature[22]), generally assumed that the effects are independent by neglecting the covariation and the interaction between drivers. Therefore, the assessment of the variations in the total indirect effect of $eCO_2$ can only be partially represented.

To address these knowledge gaps, here we investigate the dynamics in $eCO_2(ind)$ at the global scale for the period 1982–2014 using both satellite retrievals and an ensemble of Earth system models (ESMs) participating in the Coupled Model Intercomparison Project Phase 6 (CMIP6)[29] (Table 1), and project potential changes in $eCO_2(ind)$ up to the year 2100 under the SSP5-8.5 scenario. Factorial simulations derived from the fully coupled experiment and the biogeochemically coupled experiment are used to disentangle the $eCO_2(ind)$ signal for the historical and scenario periods[30] (Table 2, details in Methods). To further evaluate the robustness of model-based results, we retrieve the $eCO_2(ind)$ term from satellite observations (hereafter $eCO_2(ind)_{obs}$) through a statistical methodology within the climate analog framework (Methods). We complement the analyses by deriving $eCO_2(dir)$ through multiple non-linear regression, incorporating $CO_2$ and climate drivers, and exploring its relationship with $eCO_2(ind)$ across time and space. Finally, we investigate the sensitivity of $eCO_2(ind)$ on land aridity to elucidate the underlying eco-hydrological mechanisms.

## Results

### Temporal change in the indirect effect of $eCO_2$

An ensemble of historical simulations from seven CMIP6 models (CMIP6$_{SMA}$, SMA: simple model averaging) shows that global $eCO_2(ind)$ during the period 2000–2014 is significantly ($p < 0.05$, $t$ test) lower than that during 1982–1996 (Fig. 1a, b). Averaged across the global vegetated areas, $eCO_2(ind)$ simulated by CMIP6 models decreases from $0.24 \pm 0.32$ gC m$^{-2}$ ppm$^{-1}$ (mean $\pm$ s.e.) during 1982–1996 to $-0.04 \pm 0.24$ gC m$^{-2}$ ppm$^{-1}$ during 2000–2014 (Fig. 1a, b). The emergence of negative $eCO_2(ind)$ during 2000–2014 suggests the recent upsurge of climate stresses on the global vegetation carbon uptake, which is in agreement with the negative contribution of climate change on global GPP trend after 2000s reported in previous literature[31].

Remarkable differences in changes in $eCO_2(ind)$ emerge across geographic areas and climatological gradients. Cold and dry climate zones experience a prominent decline in $eCO_2(ind)$. The statistically significant decreasing signal is mostly in boreal regions (16.8% of global vegetated land with $p < 0.05$) with hot spots in eastern Canada, Scandinavia, and south-central Siberia (Fig. 1c, d). Warm and wet climate zones show an opposite tendency with more limited significant patterns (9.9% of global vegetated land with $p < 0.05$) (Fig. 1c, d).

In parallel, we used satellite retrievals of near-infrared reflectance of vegetation (NIRv) as a proxy of observed GPP[32] to further verify the robustness of the signals derived from model simulations. The satellite-observed $eCO_2(ind)_{obs}$ was disentangled from the other confounding effects through a climate analog approach[33], based on the identification of years with similar climate and distinct atmospheric $CO_2$ concentration (details in Methods). Observation-based results confirm a global weakening effect of $eCO_2$-driven climate change on GPP between the two periods (2000–2014 versus 1982–1996), with an overall change in $eCO_2(ind)_{obs}$ of $-0.38$ gC m$^{-2}$ ppm$^{-1}$. We also found a good agreement between model-based and observation-based results in terms of spatial patterns emerging across climatological and latitudinal gradients (Fig. 1d–f).

A comprehensive set of experiments was additionally performed to test whether our model-based results were potentially affected by the data source, temporal window length, and the criteria used to define the growing season (Supplementary Text 1 and 2; Supplementary Figs. 1 and 2, and Table 1). Meanwhile, analyses replicated by using the kernel normalized difference vegetation index (kNDVI) as an alternative satellite GPP proxy were also performed to further verify the robustness of our results (Supplementary Text 3; Supplementary Fig. 3). Altogether, these results univocally show a substantial reduction of the indirect effect of $eCO_2$ at the global scale (Fig. 1a and Supplementary Figs. 1–3) and particularly in the Northern Hemisphere (Fig. 1d–f and Supplementary Fig. 3c). Such patterns agree with the weakening temperature-vegetation relationship in northern ecosystems documented in previous literature[22], and appear plausibly influenced by the increasing water limitation (Supplementary Fig. 4).

$eCO_2(ind)$ is expected to further decline in all investigated future temporal periods under the SSP5-8.5 scenario to the point that the global mean could persistently settle on negative values (Fig. 2a). Five out of seven individual ESMs agree that $eCO_2$-driven climate change will exert a negative role on the global vegetation carbon uptake for the period 2086–2100, albeit the inter-model spread is considerable (Supplementary Fig. 5a). For the period 2086–2100, the global $eCO_2(ind)$—as estimated by CMIP6$_{SMA}$—is projected to decrease significantly by 0.36 gC m$^{-2}$ ppm$^{-1}$ compared to the analogous estimate derived for the period 1982–1996 ($p < 0.01$, $t$ test) (Fig. 2a). Such decreasing signal appears statistically significant ($p < 0.05$) over 46.5% of global vegetated land and prominently in the Northern Hemisphere (Fig. 2b,c). The global declining signal is partially dampened by opposite increasing patterns mainly occurring along the equatorial belt, which, however, manifest statistically significant over a smaller extent (32.7%).

**Table 1 | Information of CMIP6 ESMs used in this study**

| Model name | Land surface component | Modeling center | Soil depth (m) |
|---|---|---|---|
| ACCESS-ESM1-5 | CABLE2.4 with CASA-CNP | Commonwealth Scientific and Industrial Research Organisation, Australia | 2.872 |
| CanESM5 | CLASS-CTEM | Canadian Center for Climate Modeling and Analysis | 4.1 |
| CNRM-ESM2-1 | ISBA-CTRIP | Center National de Recherches Meteorologiques, France | 10 |
| E3SM-1-1 | ELM1.1 | U.S. Department of Energy | 35.18 |
| MIROC-ES2L | MATSIRO with VISIT-e | Japan Agency for Marine-Earth Science and Technology | 14 |
| MRI-ESM2-0 | HAL1.0 | Meteorological Research Institute of the Japan Meteorological Agency | 8.5 |
| UKESM1-0-LL | JULES-ES-1.0 | U.K. Natural Environment Research Council and Met Office | 2 |

**Table 2 | Description of CMIP6 factorial simulations**

| Simulation name | Type | Forcing constraints | | |
| --- | --- | --- | --- | --- |
| | | CO$_2$ radiative forcing | CO$_2$ physiological forcing | Other forcings |
| historical (1850–2014) | Fully-coupled mode | Yes, CO$_2$ increases from 285 ppm to 397 ppm | Yes, CO$_2$ increases from 285 ppm to 397 ppm | Yes, factors including CH$_4$, N$_2$O, aerosols, and land use vary over time |
| hist-bgc (1850–2014) | Biogeochemically-coupled mode | No, CO$_2$ fixed at 285 ppm (pre-industrial level) | Yes, CO$_2$ increases from 285 ppm to 397 ppm | Yes, factors including CH$_4$, N$_2$O, aerosols, and land use vary over time |
| hist-CO$_2$ (1850–2014) | Single-forcing mode | Yes, CO$_2$ increases from 285 ppm to 397 ppm | Yes, CO$_2$ increases from 285 ppm to 397 ppm | No, factors except CO$_2$ fixed at the pre-industrial level |
| ssp585 (2015–2100) | Fully coupled mode | Yes, CO$_2$ increases from 397 ppm to 1135 ppm | Yes, CO$_2$ increases from 397 ppm to 1135 ppm | Yes, factors including CH$_4$, N$_2$O, aerosols, and land use vary over time |
| ssp585-bgc (2015–2100) | Biogeochemically coupled mode | No, CO$_2$ fixed at 285 ppm (pre-industrial level) | Yes, CO$_2$ increases from 397 ppm to 1135 ppm | Yes, factors including CH$_4$, N$_2$O, aerosols, and land use vary over time |

To derive a more comprehensive picture of the terrestrial ecosystem response to eCO$_2$-driven climate change, we explored the temporal change in the strength of indirect CO$_2$ effect on carbon release by respiration (Supplementary Fig. 6), and on the net ecosystem carbon uptake (eCO$_2$(ind)-NEP, Fig. 2d, e) through factorial experiments of CMIP6 ESMs. We estimated a global eCO$_2$(ind)-NEP of −0.02 gC m$^{-2}$ ppm$^{-1}$ during the whole historical period (1982–2014), which is consistent with previous findings about the negative carbon-climate feedback from the land's perspective (i.e., positive from the atmosphere's perspective)[34]. In pace with the attenuation of the indirect effect on total vegetation carbon uptake (Fig. 2a), global eCO$_2$(ind)-NEP is projected to decrease from 0.05 ± 0.12 gC m$^{-2}$ ppm$^{-1}$ during 1982–1996 to −0.05 ± 0.03 gC m$^{-2}$ ppm$^{-1}$ during 2086–2100 (inset box in Fig. 2d). The smaller decline in eCO$_2$(ind)-NEP compared to that in eCO$_2$(ind) (−0.1 versus −0.36 gC m$^{-2}$ ppm$^{-1}$) suggests the concurrently reduced influence on ecosystem respiration and its consequent offsetting effect. The latitudinal gradient of the changes in eCO$_2$(ind)-NEP between the two periods is largely concordant with the one derived from the changes in eCO$_2$(ind), thus reflecting similar spatial dependences on environmental factors (Fig. 2b–e).

### Relationship between the indirect and direct effects of eCO$_2$

To quantify the relative importance of indirect versus direct effects of eCO$_2$ in regulating vegetation carbon uptake, simulated and observed eCO$_2$(dir) was derived based on a multiple non-linear regression (Methods) (i.e., CMIP6$_{SMA}$ and obs-RM in Fig. 3a). The analyses were complemented by two additional independent estimates of eCO$_2$(dir) based on factorial experiments of CanESM5 (i.e., CanESM5-FE in Fig. 3a), and on the climate analog approach applied to observational datasets (i.e., obs in Fig. 3a) (details in Methods). We found that along with the decrease in eCO$_2$(ind), global eCO$_2$(dir) has dropped as well in recent years and is expected to further decline in the coming decades (Fig. 3a). Model results based on factorial experiments and non-linear regression show a strong reduction in global eCO$_2$(dir) between the periods 2000–2014 and 1982–1996, largely in agreement with satellite-derived estimates (Fig. 3a). Nevertheless, the magnitude of the decline simulated by CMIP6$_{SMA}$ (−0.44 gC m$^{-2}$ ppm$^{-1}$, or −22.8%) is clearly lower than the analogous estimate derived from satellite product (obs: −1.20 gC m$^{-2}$ ppm$^{-1}$ or −78.3%; obs-RM: −1.65 gC m$^{-2}$ ppm$^{-1}$ or −67.0%) and from dedicated factorial experiments (CanESM5-FE: −1.38 gC m$^{-2}$ ppm$^{-1}$ or −69.2%) (Fig. 3a). While we recognized the intrinsic difficulties of disentangling drivers and producing robust causal attribution in observation-based analysis, we argued the emerging differences between models and observations could be partially attributable to the simplifying assumptions of CMIP6 models.

Under the investigated SSP5-8.5 scenario, the relative importance of eCO$_2$(ind) and eCO$_2$(dir) for the terrestrial carbon cycle is expected to vary greatly. The relative contribution of eCO$_2$(ind) to the net effect of eCO$_2$ (i.e., eCO$_2$(net), the sum of eCO$_2$(dir) and eCO$_2$(ind)) will likely decrease from 11.1% (1982–1996) to −22.6% (2086–12100) (Fig. 3b). On the contrary, the relative contribution of eCO2(dir) is projected to increase, mainly due to the higher relative decreasing rate of eCO$_2$(ind). However, in view of the expected progressive decline in both eCO$_2$(ind) and eCO$_2$(dir), eCO$_2$(net) could become negative, and eCO$_2$(ind) could emerge as the dominant driver of the future temporal dynamic of GPP. Some regions of the globe, such as central Canada, northern Amazon, and western and southern Africa, could exhibit a dominant role of eCO$_2$(ind) by the end of 21st century (Supplementary Fig. 7). A detailed analysis suggests that the negative eCO$_2$(ind) will overcome the positive eCO$_2$(dir) over 30.3% of global vegetated land by 2041–12055, and over 48.0% by 2086–12100 (Supplementary Fig. 8). These results agree with previous studies which have emphasized the expected net negative role of eCO$_2$ on the terrestrial carbon uptake as a result of the increasing detrimental impacts of climate change on vegetation and a saturating CO$_2$ fertilization[13].

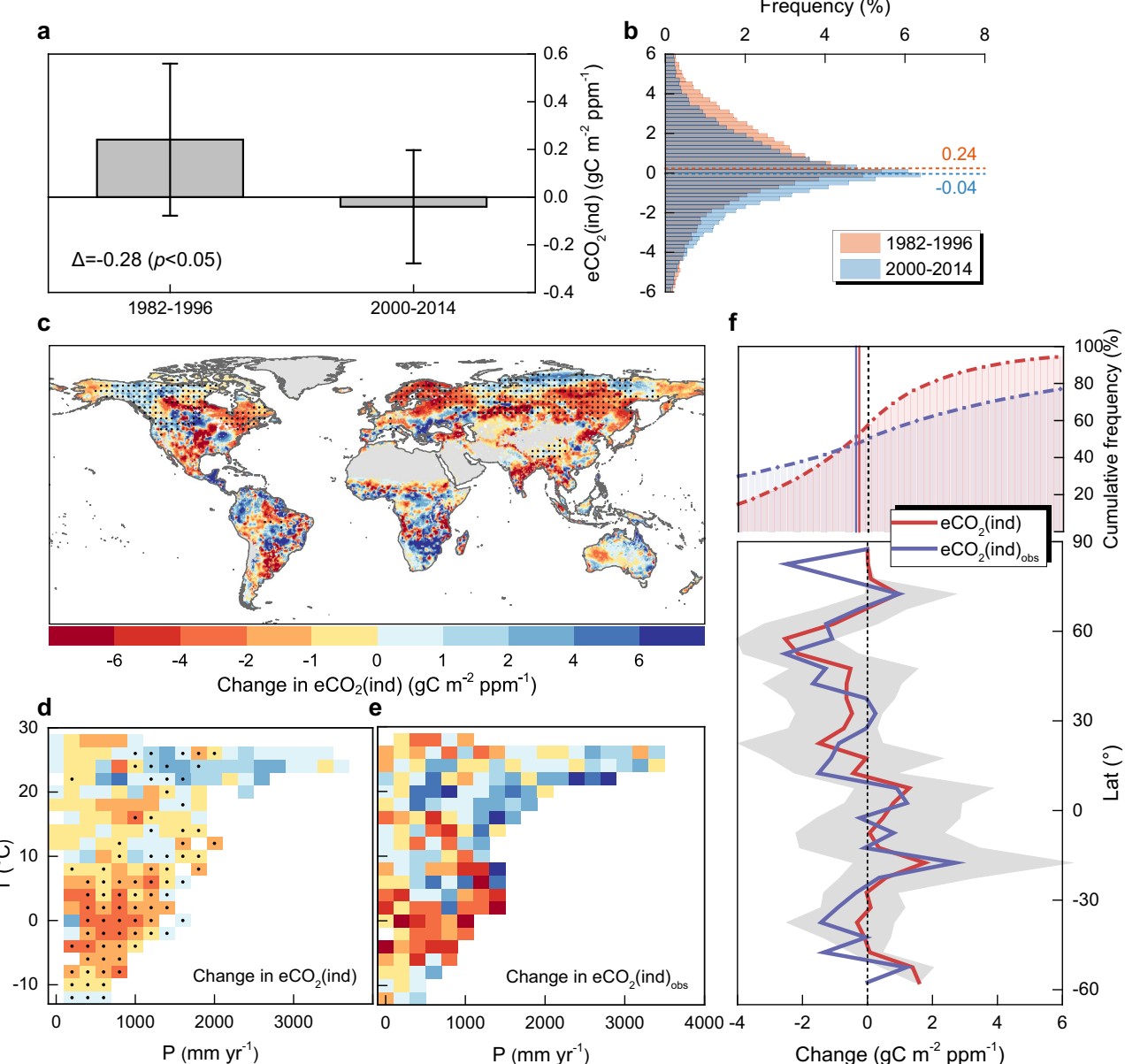

**Fig. 1 | Historical variations in the indirect effect of elevated atmospheric $CO_2$ concentration (e$CO_2$) on vegetation carbon uptake. a** Mean indirect effect of e$CO_2$ on growing-season gross primary production (GPP) via associated climate change (e$CO_2$(ind)) during the periods 1982–1996 and 2000–2014, as derived from the CMIP6 model ensemble (i.e., CMIP6$_{SMA}$). Error bars represent the standard error of effects derived from ensemble members (i.e., seven CMIP6 ESMs). $\Delta$ expresses the mean of difference in e$CO_2$(ind) between the two periods. The statistical significance of the difference is assessed by $t$ test. **b** Frequency distribution of e$CO_2$(ind) at the global scale during the periods 1982–1996 and 2000–2014, as estimated with CMIP6$_{SMA}$. Distribution averages are shown as dotted horizontal lines. **c** Spatial pattern of difference in e$CO_2$(ind) between the two periods (2000–2014 versus 1982–1996) derived from CMIP6$_{SMA}$. Non-vegetated areas are excluded from our analysis and are shown in gray. Regions labeled by black dots indicate differences that are statistically significant ($t$ test, $p < 0.05$). Dots are spaced 3° in both latitude and longitude, and statistics were computed over 9° × 9° spatial moving windows. **d** Mean difference in e$CO_2$(ind) between the two periods (2000–2014 versus 1982–1996) simulated by CMIP6$_{SMA}$, binned as a function of climatological mean precipitation (P) and air temperature (T). Black dots indicate bins with differences that are statistically significant ($t$ test, $p < 0.05$). **e** Same as **d**, but for e$CO_2$(ind)$_{obs}$ which was estimated by the satellite-observed GPP$_{obs}$ within a temporal climate analog framework. **f** (Cumulative frequency distribution of difference in e$CO_2$(ind), and e$CO_2$(ind)$_{obs}$ between the two periods (2000–2014 versus 1982–1996). Distribution averages are shown as solid vertical lines. The subplot below shows the zonal medians of difference in e$CO_2$(ind), and e$CO_2$(ind)$_{obs}$ between the two periods (2000–2014 versus 1982–1996) at 5° latitudinal resolution. Corresponding interquartile ranges of CMIP6$_{SMA}$ simulation are shown as shaded bands. Source data are provided as a Source Data file.

Results reveal that 66.9% of global vegetated land could experience the same direction of changes in e$CO_2$(ind) and e$CO_2$(dir) (i.e., "+ +" and "− −" in Fig. 3c) between the historical (1982–11996) and future (2086–2100, SSP5-8.5) period, while the remaining 33.1% could manifest reverse directions of changes (i.e., "+ −" and "− +"). The concurrent decrease in e$CO_2$(ind) and e$CO_2$(dir) ("− −") appears to be the most pervasive case (48.5%), particularly over northern latitudes

(Fig. 3c). Averaged across regions with simultaneous reductions in e$CO_2$(ind) and e$CO_2$(dir), changes in e$CO_2$(ind) and e$CO_2$(dir) explain 47.9% and 52.1% of reduction in net $CO_2$ effect on growing-season GPP (−4.87 gC m$^{-2}$ ppm$^{-1}$), respectively (Fig. 3d). Such concurrent decrease in e$CO_2$(ind) and e$CO_2$(dir) is also reflected in the sharper decrease in e$CO_2$(net) in northern lands (−2.69 gC m$^{-2}$ ppm$^{-1}$, or −82.0%) between the historical and future periods compared to the global mean

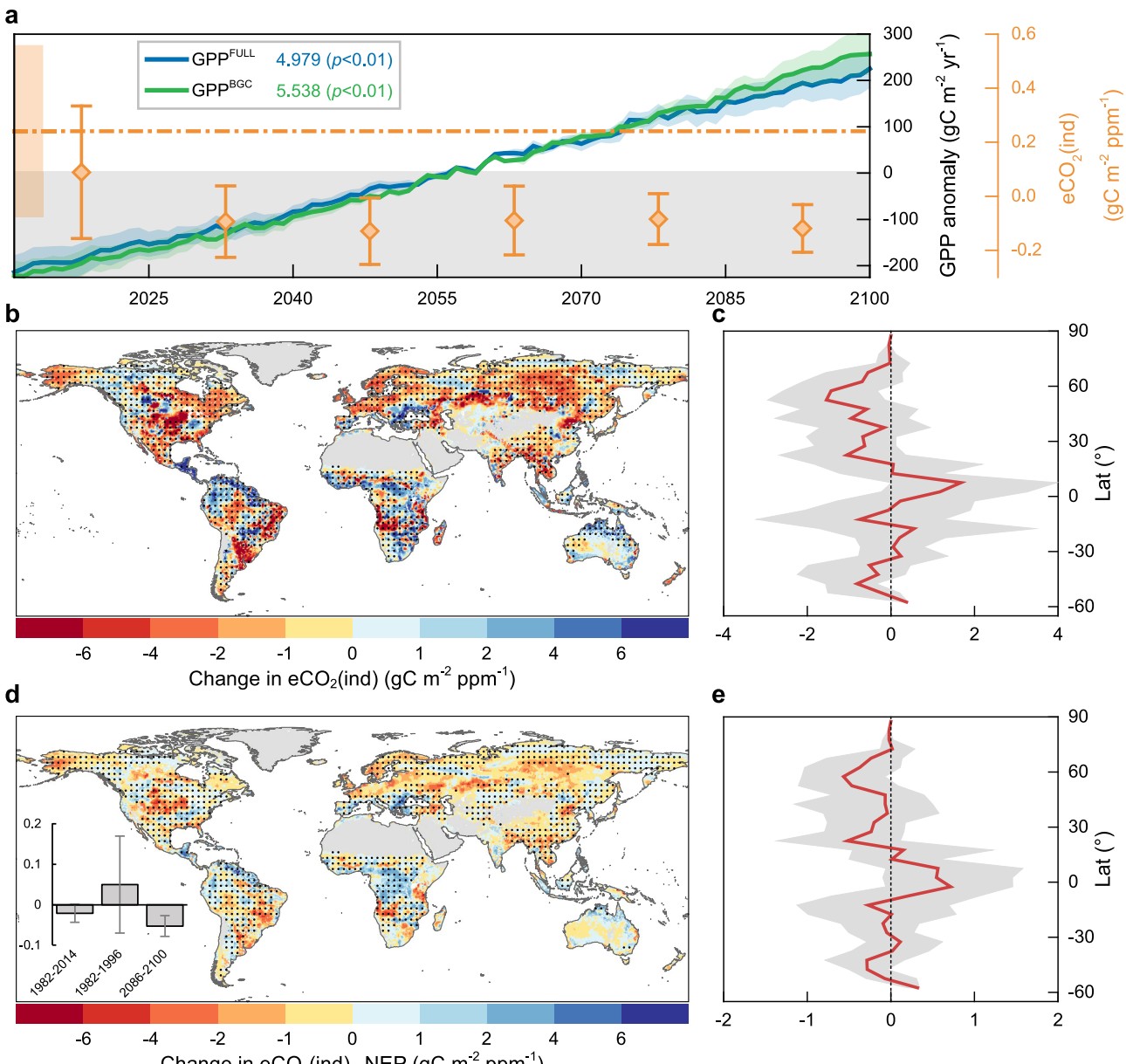

**Fig. 2 | Projection of future variations in indirect effect of elevated atmospheric CO₂ concentration (eCO₂) on vegetation carbon uptake. a** Mean indirect effect of eCO₂ on growing-season gross primary production (GPP) via associated climate change (eCO₂(ind)) derived from CMIP6$_{SMA}$ during the six independent periods, namely 2011–2025, 2026–2040, 2041–2055, 2056–2070, 2071–2085, and 2086–2100. Ensemble mean and standard error are shown by the diamond symbol and whiskers, both referring to the right orange *y* axis. Dotted horizontal line and its shaded band represent the eCO₂(ind) during 1982–1996 and the corresponding standard error, as estimated by CMIP6$_{SMA}$. Interannual changes in anomalies of growing-season GPP over 2011–2100 globally, simulated by CMIP6$_{SMA}$ under the fully-coupled experiment (GPP$^{FULL}$) and the biogeochemically-coupled experiment (GPP$^{BGC}$) are shown in blue and green lines, respectively. Numbers refer to the trends of GPP$^{FULL}$ and GPP$^{BGC}$ (unit: gC m² yr⁻²) over 2011–2100. The statistical significance of trends is assessed by Mann–Kendall test. **b** Spatial pattern of difference

in eCO₂(ind) between the historical and future periods (2086–2100 versus 1982–1996) derived from CMIP6$_{SMA}$. Regions labeled by black dots indicate differences that are statistically significant (*t* test, $p < 0.05$). Dots are spaced 3° in both latitude and longitude, and statistics were computed over 9°×9° spatial moving windows. **c** Zonal medians of difference in eCO₂(ind) between the historical and future periods (2086–2100 versus 1982–1996) simulated by CMIP6$_{SMA}$ at 5° latitudinal resolution. Corresponding interquartile ranges of CMIP6$_{SMA}$ simulation are shown as shaded band. **d, e** Same as **b, c** but for the indirect effect of eCO₂ on growing-season net ecosystem production (NEP) via associated climate change (eCO₂(ind)-NEP) derived from CMIP6$_{SMA}$. The inset in **d** shows the mean eCO₂(ind)-NEP during the periods 1982–2014, 1982–1996, and 2086–2100, respectively. Error bars represent the standard error of effects derived from ensemble members. Source data are provided as a Source Data file.

(−1.65 gC m⁻² ppm⁻¹, or −75.7%), as estimated by CMIP6$_{SMA}$ (Supplementary Table 2).

Additional analyses based on model simulations from the idealized 1% per year increasing CO₂ experiments show spatial patterns and trends in both indirect and direct CO₂ effects at the global mean level similar to those described above (Supplementary Text 4;

Supplementary Figs. 9–11). Meanwhile, under such idealized scenario where radiatively-coupled mode is available, estimates of direct CO₂ effect based on the non-linear regression framework and those based on the climate analog approach were carefully compared against those obtained directly from factorial experiments (Supplementary Text 4). The high agreement among the three sets of estimates further

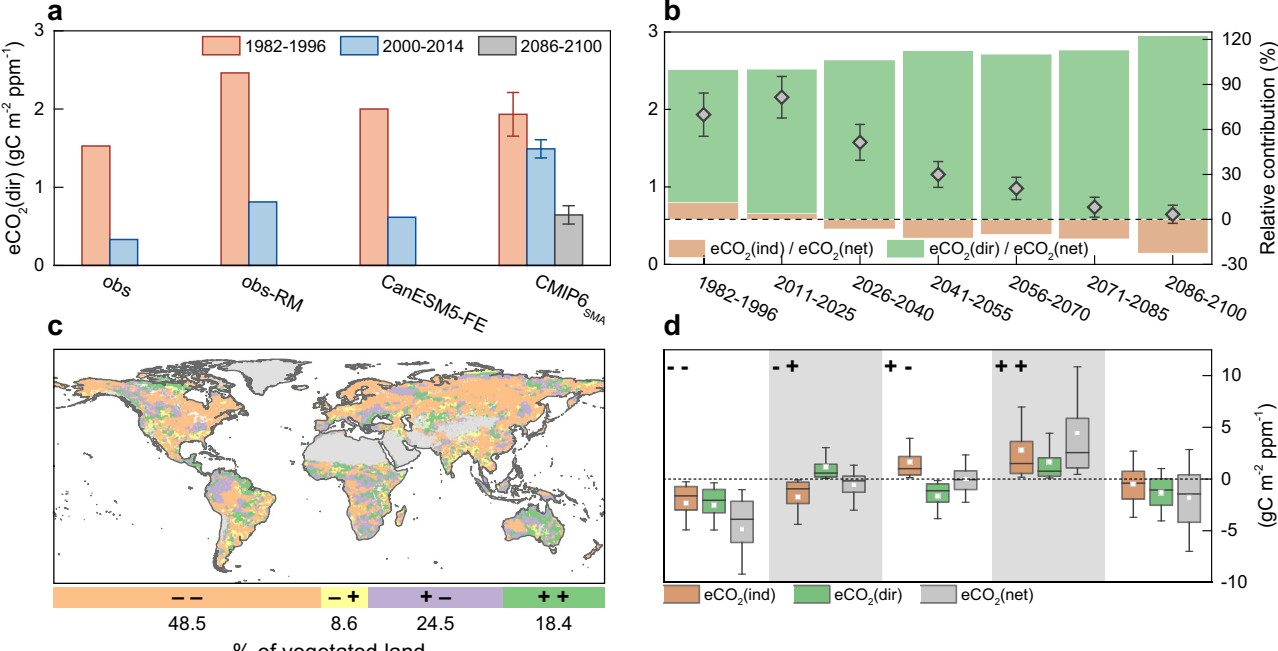

**Fig. 3 | Relationship between direct and indirect effects of elevated atmospheric CO₂ concentration (eCO₂) on vegetation carbon uptake. a** Mean direct physiological effect of eCO₂ on growing-season gross primary production (GPP) (eCO₂(dir)) during periods 1982–1996 and 2000–2014, simulated by CanESM5 factorial experiments (i.e., CanESM5-FE), and estimated by observed GPP under the temporal climate analog framework (i.e., obs), estimated by observed GPP in combination with the non-linear regression model (i.e., obs-RM), and estimated by CMIP6$_{SMA}$-simulated GPP in combination with the non-linear regression model (i.e., CMIP6$_{SMA}$). Mean eCO₂(dir) during the period 2086–2100 under SSP5-8.5 projected by CMIP6$_{SMA}$ is also provided. Error bars represent the standard error of effects derived from ensemble members (i.e., seven CMIP6 ESMs). **b** Mean eCO₂(dir) derived from CMIP6$_{SMA}$ and its standard error during seven independent periods, namely 1982–1996, 2011–2025, 2026–2040, 2041–2055, 2056–2070, 2071–2085, and 2086–2100, shown by diamond symbol and whiskers. Bars in green

and orange represent the relative contributions of the indirect effect of eCO₂ (eCO₂(ind)) and eCO₂(dir) to the net effect of eCO₂ (eCO₂(net)) during corresponding periods and derived from CMIP6$_{SMA}$. **c** Spatial pattern of relationship between changes in eCO₂(ind) and eCO₂(dir) between historical and future periods (2086–2100 versus 1982–1996), where "– –" represents decrease in eCO₂(ind) and decrease in eCO₂(dir), "– +" represents decrease in eCO₂(ind) and increase in eCO₂(dir) and so on. The legend shows the fraction of vegetated areas for each thematic class (i.e., "– –", "– +", "+ –", and "+ +"). **d** Boxplot of changes in eCO₂(ind), eCO₂(dir), and eCO₂(net) between historical and future periods (2086–2100 versus 1982–1996) for different thematic classes mentioned in **c** and for the globe (rightmost). Boxplot elements: box = values of 25th and 75th percentiles; horizontal line = median; rectangle = mean; whiskers = values of 10th and 90th percentiles. Source data are provided as a Source Data file.

supports the validity of our multiple non-linear regression framework and of the climate analog approach (Supplementary Figs. 12 and 13).

GPP in the Northern Hemisphere has increased steadily during the past decades[35] in response to the large and positive eCO₂(ind) and eCO₂(dir) (Supplementary Fig. 14), thus playing a critical contribution to the global terrestrial carbon sink[36,37]. Therefore, the widespread and strong decrease in both indirect and direct effects of eCO₂ in the Northern Hemisphere resulting from our analyses rises concern about the future dynamic of the regional carbon sink and its capacity to keep the pace of anthropogenic emissions.

**Mechanisms behind the decline in the indirect effect of eCO₂**
To disentangle the possible mechanisms responsible for the declining eCO₂(ind), we explored its relationship with the expected changes in terrestrial water availability. To this aim, we first exploited the CMIP6 simulations to quantify the spatiotemporal variations in aridity conditions, here expressed in terms of surface (0–10 cm) soil moisture (SM$_{surf}$). Results indicate a projected widespread decline in terrestrial water availability by the end of the century compared to the current conditions (82.6% of global vegetated land exposed to a reduction in SM$_{surf}$, Fig. 4a). At the global level and based on multi-model means (i.e., CMIP6$_{SMA}$), we estimated a significant decrease in SM$_{surf}$ during 2086–2100 by 7.3% ($p < 0.01$, $t$-test) compared to analogous estimates obtained for the period 1982–1996 (Fig. 4b). Similar drying patterns emerge for individual model runs (Supplementary Fig. 15), for total soil moisture (SM$_{total}$), for a widely used aridity index (defined as the ratio

of annual precipitation to potential evapotranspiration, P/PET) (Supplementary Figs. 16 and 17). Previous studies focusing on dryness indices[38] and hydrological regimes[39,40] further corroborate such drying trends.

To investigate the relationship between change in eCO₂(ind) and land surface drying/wetting, we averaged the change in eCO₂(ind) across gradients of mean annual SM$_{surf}$ during 1982–1996 and the corresponding change in SM$_{surf}$ (i.e., 2000–2014 versus 1982–1996, and 2086–2100 versus 1982–1996). SM$_{surf}$ = 0.26 m³ m⁻³ generally corresponds to P/PET = 1 at the mean annual scale (Supplementary Fig. 17b), which is widely treated as the threshold between non-humid and humid regions[41,42]. We found that eCO₂(ind) generally declines (enhances) with the land drying (wetting) in humid regions (SM$_{surf}$ > 0.26 m³ m⁻³, Supplementary Fig. 4a) in both historical and scenario simulations (Fig. 4c, d). However, in water-limited conditions (SM$_{surf}$ < 0.26 m³ m⁻³), the weakened negative eCO₂(ind) along with the land drying results in a negative relationship between changes in eCO₂(ind) and SM$_{surf}$ (Fig. 4c, d and Supplementary Fig. 7). CO₂ and drought-related enhancement in growing-season water-use efficiency (WUE) (Supplementary Fig. 18), relax the water limitation to vegetation growth, especially over semi-arid climate zones[24,43–45], and may consequently limit the negative trend in eCO₂(ind) (Fig. 4d). In addition, for water-limited environments, a decrease in eCO₂(ind) occurs consistently under both land drying and wetting, indicating the possible importance of other factors, such as vegetation type and species diversity, in modulating the vegetation response to climate change.

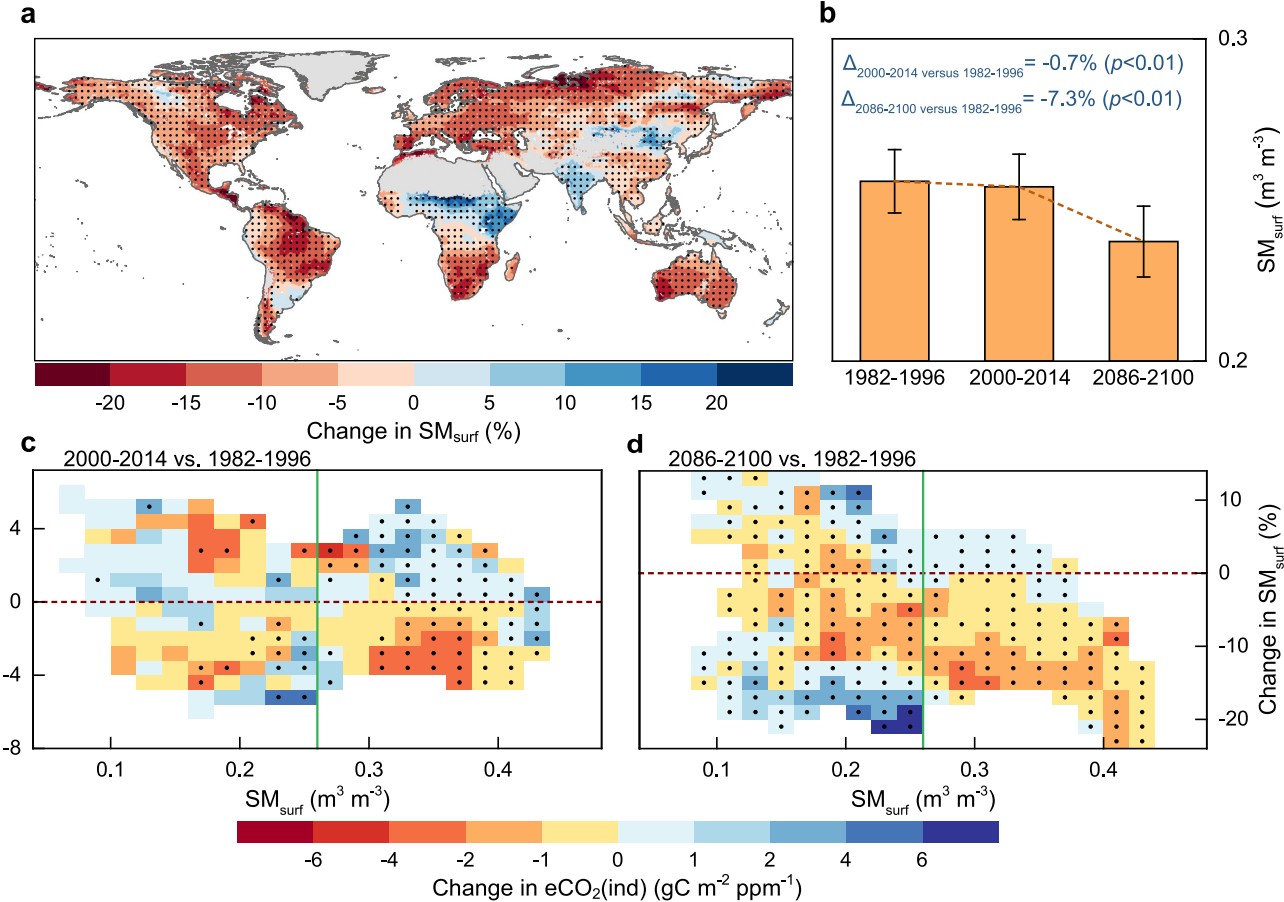

**Fig. 4 | Sensitivity of indirect CO$_2$ effect on terrestrial water availability.**
**a** Spatial pattern of relative change in surface soil moisture (SM$_{surf}$) between the historical and future periods (2086–2100 versus 1982–1996) derived from CMIP6$_{SMA}$. Regions labeled by black dots indicate changes that are statistically significant (t test, $p < 0.05$). Dots are spaced 3° in both latitude and longitude, and statistics were computed over 9° × 9° spatial moving windows. **b** Global mean SM$_{surf}$ derived from CMIP6$_{SMA}$ during the period 1982–1996, 2000–2014, and 2086–2100, respectively. Numbers refer to change in SM$_{surf}$ relative to 1982–2016. **c** Difference in indirect effect of elevated atmospheric CO$_2$ concentration (eCO$_2$) on growing-season gross primary production (GPP) via associated climate change

(eCO$_2$(ind)) between the periods 1982–1996 and 2000–2014 derived from CMIP6$_{SMA}$, binned as a function of corresponding changes in SM$_{surf}$ and mean annual SM$_{surf}$ (Supplementary Fig. 4a). SM$_{surf}$ = 0.26 m$^3$ m$^{-3}$ (i.e., the green solid line) overall corresponds to the ratio of annual precipitation to potential evapotranspiration (P/PET) = 1 at the mean annual scale, that is the threshold between non-humid and humid regions (Supplementary Fig. 17b). Black dots indicate bins with differences that are statistically significant (t test, $p < 0.05$). **d** Same as **c**, but for the difference between the periods 1982–1996 and 2086–2100. Source data are provided as a Source Data file.

Similar sensitivities to increasing water limitation have been obtained using SM$_{total}$ and P/PET in place of SM$_{surf}$ and referring to different temporal window lengths (Supplementary Figs. 16, 17, and 19). Such high consistency demonstrates the substantial independence of our results on the proxy of terrestrial water availability and the selection of time-window length.

## Discussion
Our study provides multiple and coherent evidence that the indirect effect of eCO$_2$ on global vegetation carbon uptake via associated climate change has declined over the last three decades (Fig. 1 and Supplementary Fig. 3). The signal of the ongoing trends has been derived both from Earth system models' simulations and from satellite observations using a statistical approach to disentangle direct and indirect CO$_2$ effects from time series analysis. In addition, results show that the positive indirect effect of eCO$_2$ that has stimulated the global GPP in recent years will most likely continue to decline in the future, particularly in northern high latitudes, and turn into negative values firmly under the high CO$_2$ emission scenario (Fig. 2). This epochal change in the sign of the indirect CO$_2$ effect may lead to a positive land carbon-climate feedback from the atmosphere's perspective[34,46].

The interpretation of the CO$_2$ effects mediated by climate change is intrinsically complex because primary productivity is controlled by different environmental drivers in the different biomes, like low temperature in the boreal regions, incoming radiation in the humid tropics, or water availability in the arid regions[27]. For this reason, analyses have to address multiple factors and their interactions at once[28,47]. However, previous assessments were largely based on the analysis of a single climatic factor (e.g., temperature[22], water availability[48], and VPD[23]), while our study presents an attempt to integrate multiple drivers across the different World regions. For instance, the strong signal emerging in the boreal regions (Figs. 1f and 2c) can be partially attributable to the weakening temperature-vegetation relationship in northern ecosystems[22], which may be related to the nonlinear response of photosynthesis to temperature, increased extreme heat, increasing water limitation driven by the anticipation of phenology[49] and expansion of woody shrubs[50,51].

Our assessment shows that the increasing water limitation is a critical driver of the weakened indirect effect of eCO$_2$ on the global vegetation carbon uptake (Fig. 4 and Supplementary Figs. 16, 17 and 19). This phenomena may be driven by the detrimental effect of water

scarcity on the resilience of global vegetation to climate variability and extremes[52]. An exception to this pattern occurs in water-limited environments, where the negative indirect effect of $eCO_2$ seems to weaken with the land drying (Fig. 4c, d), probably due to the enhancement of direct $CO_2$ effect on WUE and the resulting mitigating effect on water constraints[24] (Supplementary Fig. 18). A recent analysis focusing on global forests suggests that WUE and aridity index are closely and negatively related below a threshold value of aridity index≈1[53], partly supporting our finding.

The complex interplay between the rapid changes in climate conditions and the increasing risks of natural disturbances may also contribute to the transition from positive to negative indirect $CO_2$ effects. For example, warmer and drier conditions facilitate insect outbreaks, while warmer and wetter conditions increase disturbance from pathogens[54]. The expected intensification of disturbances may amplify the negative effect of climate change on primary productivity by enhancing plant vulnerability and mortality rates[55]. However, the abovementioned processes have not yet been fully considered in state-of-the-art dynamic vegetation models and ESMs that are used in major climate assessments like the Intergovernmental Panel on Climate Change (IPCC) Assessment Reports[56,57]. The simplified representation of disturbances and mortality in these models may ultimately hamper our full understanding of the ongoing and future variations in carbon-climate feedback, and in turn, lead to the overestimation of the future terrestrial carbon sink due to the misestimation of the indirect effect of $eCO_2$. In fact, our study shows that CMIP6 model simulations clearly report a lower magnitude of the decline in global indirect $CO_2$ effect during the historical period compared with observation-based estimates (Fig. 1f). This could be attributable to the poor representation of water limitation and natural disturbances in current ESMs, the limited consideration of vegetation mortality associated with biotic agents and the non-linearity of the processes behind[57,58]. Considering that disturbance regimes are expected to intensify in many parts of the globe because of climate change[54,58,59], the enhanced representation of the phenomena in ESMs would plausibly produce an even stronger decline in the global projection of the indirect $CO_2$ effect.

The weakened indirect effect of $eCO_2$ reported here, in addition to the concurrent decline in direct physiological effect, confirms the transitory nature of the strong growth benefits induced by climate warming and $CO_2$ fertilization, especially in the boreal regions[9,60]. These findings are in accordance with the expectation about the future saturation of $CO_2$ fertilization effect and the increasingly negative effect of climate change on vegetation[13]. More importantly, the simultaneous reductions in indirect and direct effects of $eCO_2$ imply that $eCO_2$ may exert a less positive up to negative role on the terrestrial carbon uptake in the future (Fig. 3), consequently reducing the ecosystems' capacity to sequester atmospheric $CO_2$. These phenomena may ultimately lead to an acceleration of climate change in the second part of the century, further challenging the efforts of humanity toward carbon neutrality. In addition, more frequent and severe climate extremes in a warming climate[61,62], e.g., increasing drought conditions, may further aggravate the decline in the indirect effect of $eCO_2$ (Fig. 4) as well as that in direct $CO_2$ fertilization[63]. The intensification of a positive feedback loop between climate change and land $CO_2$ emission undoubtedly would limit the potential of terrestrial ecosystems to serve as carbon sinks and have great implications for the efficacy of land-based mitigation policies and for the societal efforts required for meeting climate mitigation targets. In this respect, our results contribute to a better understanding of global change impacts on terrestrial ecosystems under current and future conditions, and meanwhile, may help the development of more integrated and realistic mitigation strategies, by informing climate policies on the weakening of the fertilization effects of $eCO_2$ and associated amplification of climate warming.

## Methods

### CMIP6 simulations

To explore the indirect effect of $eCO_2$ on vegetation carbon uptake, we used outputs from an ensemble of seven Earth system models (ESMs) that participate in the carbon-climate feedback experiment (C4MIP) within the framework of the Coupled Model Intercomparison Project Phase 6 (CMIP6)[30] (https://esgf-node.llnl.gov/search/cmip6/): ACCESS-ESM1-5, CanESM5, CNRM-ESM2-1, E3SM-1-1, MIROC-ES2L, MRI-ESM2-0, and UKESM1-0-LL (Table 1). These models were selected because they provide simulations under different coupling modes required to disentangle the effects of $eCO_2$. We focused on the SSP5-8.5 scenario because C4MIP simulations are available only for the highest emission trajectory ($CO_2$ concentration is projected to reach 1135 ppm in 2100[64]). The ESMs have full carbon cycles, which include carbon uptake by vegetation that varies in response to changes in atmospheric $CO_2$ concentration and climate[46]. For each ESM, one biogeochemically-coupled experiment and one fully-coupled experiment in both historical (1982–2014) and future scenario (2015–2100) periods were analyzed within a factorial simulation framework (Table 2). In biogeochemically coupled experiments ("hist-bgc" and "ssp585-bgc" in the CMIP6 terminology), biogeochemical processes over land respond to $eCO_2$, whereas the radiative code experiences fixed $CO_2$. In the fully-coupled experiment ("historical" and "ssp585" in the CMIP6 terminology), both radiative and biogeochemical processes respond to $eCO_2$ (consistent with observations in the historical period). All other forcings (e.g., non-$CO_2$ greenhouse gases, aerosols, and land use) are identical for these two sets of experiments, i.e., time-varying in both radiative and biogeochemical processes. Furthermore, we used outputs from $CO_2$ individual forcing experiment ("hist-$CO_2$" in the CMIP6 terminology) conducted by the Detection and Attribution Model Intercomparison Project (DAMIP)[65] (Table 2). "hist-$CO_2$" experiment refers to the historical simulation driven only by observed changes in $CO_2$ concentration, with other forcings keeping temporally constant (e.g., non-$CO_2$ greenhouses gases, aerosols, and land use). Combining "hist-$CO_2$" with "historical" and "hist-bgc" enables to quantify the historical $eCO_2$(dir) through factorial simulations of CMIP6 runs. Unfortunately, $CO_2$ individual forcing experiment has not yet been extended to the future period, and only one ESM (i.e., CanESM5) took part in all three experiments. Because of these disadvantages, the abovementioned analysis applied for the estimation of global direct effects of $eCO_2$ was complemented by a more general regression framework extendible to the full ESM ensemble and to the different temporal periods (details in section "Quantifying indirect and direct effects of $eCO_2$ by model outputs"). Factorial simulations based on the abovementioned experiments ("hist-bgc", "historical", and "hist-$CO_2$") were elaborated in the following sections.

A set of variables generated by ESM simulations were used for the following analyses, including: monthly scale gross primary production (GPP), net primary production (NPP), autotrophic respiration ($R_a$), heterotrophic respiration ($R_h$), evapotranspiration (ET), maximum, minimum and mean air temperature ($T_{max}$, $T_{min}$, and $T$), precipitation (P), cloud cover (CL), relative humidity (RH), surface (0–10 cm) soil moisture ($SM_{surf}$, 0–10 cm), and total soil moisture ($SM_{total}$, depth depending on models, see Table 1). Considering that the hydrologically active soil depth varies greatly among the models (from 2 m in UKESM1-0-LL to 35.18 m in E3SM-1-1), $SM_{total}$ as well as $SM_{surf}$ was converted from the original gravimetric unit (kg m$^{-2}$) to volumetric unit (m$^3$ m$^{-3}$) by dividing the gravimetric soil water content by the corresponding soil depth, Such conversion allows for the comparison of results obtained from different models and the development of more robust multi-model ensembles of soil moisture. Variations in $SM_{total}$ and $SM_{surf}$ were expressed in relative terms (%) with respect to their average values computed for the baseline period (e.g., 1982–1996)[39] (Fig. 4 and Supplementary Figs. 4, 15 and 16). Only for the ES3M-1-1 model, some of the abovementioned variables were not

provided (NPP, $T_{max}$, $T_{min}$, and RH), and therefore they were retrieved by empirical formula and statistical approach (additional details reported in Supplementary Text 5). We additionally derived ecosystem respiration ($R_{eco}$) as the sum of $R_a$ and $R_h$, and the net ecosystem production (NEP) was calculated as the difference between GPP and $R_{eco}$[66]. Vapor pressure deficit (VPD), which directly relates to atmospheric water demand[19,67], was calculated based on Abbott and Tabony[68] for each grid-cell as follows:

$$VPD = 0.6108 e^{\frac{17.27T}{T+237.3}} \left(1 - \frac{RH}{100}\right) \tag{1}$$

where $T$ is given in °C, and the resulting VPD is in kPa. Furthermore, we used the ratio of mean annual P to potential evapotranspiration (PET) as the aridity index retrieved from the FAO Penman-Monteith algorithm[69]. Details on the PET estimation are reported in Supplementary Text 6.

CMIP6 outputs were resampled to a common 0.5° × 0.5° global grid-cell using the bilinear method of interpolation. Moreover, for each temporal window (e.g., 1982–1996, 2000–2014, 2086–2100, and 1982–2014), we computed the associated multi-year mean growing season at the grid-cell scale. The growing season was defined as the period spanning months with average $T > 0$°C[14] and—limitedly to arid and semi-arid ecosystems—cumulative P between 10% and 90% of the annual total P (Supplementary Fig. 20). The integration of a P threshold in the definition of the growing season for water-limited environments enables to account for possible inactive vegetation phase at $T > 0$ °C due to water deficit conditions[70]. Areas characterized by an aridity index, quantified in terms of P/PET, <1 were labeled as arid and semi-arid ecosystems[9]. For arid and semi-arid grid cells located in the Southern Hemisphere, P accumulation was set to start in July and end in June of the next year. The resulting growing season was used as a reference period to aggregate the original monthly variables provided by CMIP6 to the growing-season scale. The robustness of our results is tested with respect to two alternative definitions of the growing season period: (1) $T > 5$ °C and cumulative P between 10% and 90% of the annual total P; (2) $T > 5$°C and cumulative $P$ between 20% and 80% of the annual total $P$ (Supplementary Fig. 2). In both cases, the P threshold is applied to arid and semi-arid regions only.

## Observation-based products

We exploited the long-term GPP dataset (hereafter $GPP_{obs}$) based on near-infrared reflectance of vegetation (NIRv) retrieved from the Advanced Very High Resolution Radiometer (AVHRR) reflectance observations[32,71] (https://data.tpdc.ac.cn/en/data/d6dff40f-5dbd-4f2d-ac96-55827ab93cc5/). The satellite GPP dataset, provided at monthly temporal resolution and at 0.05° spatial resolution, has global coverage and spans the period 1982–2014 (Supplementary Fig. 21a). It has been largely validated in previous studies against ground measurements and compared with estimates derived from machine-learning upscaling approaches, light-use-efficiency models and processed-based models[32,72]. To match the spatial and temporal resolution of ESMs output, satellite GPP data were resampled to 0.5° and integrated over the growing season derived from the CMIP6 simulations in the fully-coupled experiment, as described above, to increase consistency in the data-model comparison (Supplementary Fig. 20). The obtained satellite-based growing season GPP data were used to evaluate the ESMs performance in capturing global GPP dynamics (Supplementary Figs. 22 and 23).

Furthermore, to explore the observed impact of $eCO_2$ on vegetation carbon uptake (details in the following sections), we used monthly $T_{max}$, $T_{min}$, $T$, $P$, CL, actual water vapor (VP), and PET retrieved from the Climatic Research Unit (CRU v4.05) climate dataset[73] (https://catalogue.ceda.ac.uk/) delivered for the whole globe at 0.5° spatial resolution and covering the period 1982–2014.

We additionally derived monthly VPD values as the difference between the saturated vapor pressure (SVP) and VP for each grid-cell based on the following formulation:

$$VPD = 0.6108 e^{\frac{17.27T}{T+237.3}} - VP \tag{2}$$

where $T$ is given in °C, VP and VPD are in kPa. Here we applied Eq. (2) instead of Eq. (1) to estimate VPD because RH is not available from the CRU dataset. All climatic factors were then aggregated at the growing-season temporal resolution.

We derived a global vegetated land mask from the annual land cover maps of the European Space Agency's Climate Change Initiative[74] (https://www.esa-landcover-cci.org) acquired for the period 1992–2014 at 300 m spatial resolution, referring to a simplified aggregation scheme based on physiognomy alone. Land cover maps were resampled to 0.5° using the majority method to match the common spatial resolution. All grid cells (0.5 × 0.5° resolution) classified as vegetation class (including forest, grassland, shrubland, cropland, and wetland) throughout the 23 years were defined as vegetated areas and included in our analyses (Supplementary Fig. 24).

## Assessing indirect/direct effects of eCO₂ from model outputs

Following similar approaches reported in literature[10,16,75,76], the effect of $eCO_2$-induced climate change on growing-season GPP (i.e., the indirect effect of $eCO_2$, expressed as $eCO_2(ind)$) was derived from factorial simulations of multiple CMIP6 experiments by calculating the difference between the trend in growing-season GPP generated in the fully-coupled mode and that in the biogeochemically-coupled mode normalized by the increase rate of atmospheric $CO_2$ concentration:

$$eCO_2(ind) = \frac{\delta GPP^{FULL} - \delta GPP^{BGC}}{\delta CO_2} \tag{3}$$

where $\delta GPP^{FULL}$ and $\delta GPP^{BGC}$ are the trends in growing-season GPP in the fully-coupled experiment (i.e., "historical" and "ssp585") and the biogeochemically-coupled experiment (i.e., "hist-bgc" and "ssp585-bgc"), respectively; $\delta CO_2$ represents the trend in atmospheric $CO_2$ concentration and is prescribed by CMIP6[64,77]. The statistical significance of the trends was evaluated using the nonparametric Mann–Kendall test. The absolute signal (term $\delta GPP^{FULL} - \delta GPP^{BGC}$) was normalized to the unit of gC m$^{-2}$ ppm$^{-1}$, to eliminate the impact of the difference in increasing rate of atmospheric $CO_2$ concentration in various periods (e.g., 1982–1996, 2000–2014, and 2086–2100). The term $eCO_2(ind)$ excludes the direct physiological effect of $eCO_2$ and the effects of non-$CO_2$ forcing agents on GPP, as these components have been removed from the factorial simulations (Eq. (3)). The approach enabled us to separately quantify the $eCO_2(ind)$ for different reference temporal period (e.g., 1982–1996, 2000–2014, and 2086–2100) at grid-cell level. For global-scale $eCO_2(ind)$ estimates, the GPP terms reported in (Eq. (3)) were obtained by spatial average weighting each grid-cell value based on its area (Fig. 1a). The same methodology is applied consistently for all global-scale and regional-scale aggregated metrics described in the following sections. The analyses were complemented by applying Eq. (3) to NPP, $R_a$, $R_h$, $R_{eco}$, and NEP (in place of GPP) to comprehensively evaluate the response of distinct carbon fluxes to $CO_2$ radiative forcing (Fig. 2d, e and Supplementary Fig. 6).

We quantified the direct effect of $eCO_2$ on growing-season GPP (i.e., $eCO_2(dir)$) within a multiple non-linear regression framework applied to simulations obtained from the CMIP6 fully-coupled experiment. Such approach was specifically designed since radiatively-coupled experiments ideally required to derived $eCO_2(dir)$ from factorial simulations were not available. To derive robust fitting functions, we first performed a collinearity test based on the variance inflation factor (VIF), to preliminary select what drivers to include in

the multiple regression. Results show that $CO_2$, $T_{min}$, P, VPD, and CL, show no/weak collinearity (VIF < 10) in most parts of the globe (CRU: 97.0 ∼ 100%; $CMIP_{SMA}$: 86.7 ∼ 100%) and thus were all retained in the predictor set (Supplementary Fig. 25). Considering that climate exerts non-linear control on terrestrial carbon uptake[78,79], non-linear terms (e.g., interaction and quadratic terms) were incorporated into the regression model in addition to linear terms. Following the modeling framework described in Chen et al.[80], we used stepwise regression at the grid-cell scale to reduce redundant predictors. The model form most often identified across all grid cells was ultimately adopted to each grid-cell, enabling a consistent analysis at the global scale. The adopted model is described by the following equation:

$$GPP = \beta(CO_2) + C_1(P) + C_2(VPD) + C_3(T_{min} \cdot VPD) + C_4(P \cdot CL) + C_5 + \varepsilon \quad (4)$$

where $\beta$, $C_1$, $C_2$, $C_3$, $C_4$, and $C_5$ represent the regression coefficients, and $\varepsilon$ is the residual error term. Therein, $\beta$ (gC m$^{-2}$ ppm$^{-1}$) refers to the sensitivity of GPP to $CO_2$, and thus reflects the term $eCO_2$(dir). Such an approach enabled us to disentangle the direct physiological effect of $eCO_2$ on GPP by factoring out the potentially confounding effects of climatic factors. All variables in Eq. (4) were taken from CMIP6 model simulations under "historical" and "ssp585" experiments. Regressions were estimated on annual anomalies (i.e., annual values minus the mean signal for a given period) for all variables, thus removing the background effects on vegetation but preserving those originating from interannual variations[9]. We performed an additional set of modeling experiments to test the model sensitivity on different hydrological variables. To this aim, we expressed the interannual variations in GPP within a non-linear regression that incorporates soil moisture in place of P (details in Supplementary Text 7). Test results based on the Akaike Information Criterion (AIC), the corrected Akaike Information Criterion (AICc), and the Bayesian Information Criterion (BIC) suggest that non-linear regression based on soil moisture has no substantial improvement in model performance (Supplementary Fig. 26) and leads to larger inter-model spread compared to the original one (i.e., Eq. (4)) (Supplementary Fig. 27 and Table 3). We therefore retained the regression framework based on P as defined in Eq. (4) for subsequent analyses.

For CanESM5, we estimated $eCO_2$(dir) by the use of an alternative method based on the outputs from three sets of factorial experiments available for the historical period as follows:

$$eCO_2(dir) = \frac{\delta GPP^{CO2} - (\delta GPP^{FULL} - \delta GPP^{BGC})}{\delta CO_2} \quad (5)$$

where $\delta GPP^{CO2}$ is the trend in growing-season GPP in the $CO_2$ individual forcing experiment ("hist-$CO_2$"). Estimates based on this approach were compared against those generated by the above-mentioned regression model to test the robustness of Eq. (4) in quantifying global mean $eCO_2$(dir) and its change (Fig. 3a).

## Deriving indirect effect of $eCO_2$ from observations

To further corroborate our model-based findings, we investigated the direct and indirect components of $eCO_2$ and their changes through an observation-based approach. The method is based on the pixel-level assessment of the changes in GPP under similar climate conditions but different atmospheric $CO_2$ concentrations. For this purpose, we used the climate analog approach[33] to identify couples of years with different atmospheric $CO_2$ concentrations but similar climate conditions (i.e., climate analogous (CA) years) in each time period (e.g., 1982–1996, and 2000–2014) based on the CRU v4.05 climate dataset. Temporal climate analogs are derived from the Mahalanobis distance, which is a multivariate distance independent of the scale of the climate variables[81]. For the period 1982–1996, as an example, we first identified the two sub-periods 1982–1988 and 1989–1996, and then identified for

each grid-cell the years in which the climate condition is most similar between the two sub-periods. We calculated the Mahalanobis distance based on eleven climate variables derived from the CRU v4.05 dataset and identified as key determinants for climate analog analysis in previous studies[33,81–83]. The selected variables include: mean annual CL, mean annual VPD, mean annual T, total annual P, annual P/PET, mean T, and total P for December–February (DJF) and June-August (JJA), T seasonality (represented by the standard deviation of monthly T), and P seasonality (represented by the coefficient of variation in monthly P). In order to reduce the dimensionality of the data space, we applied a principal component analysis to all climate variables and discarded the principal components with variance <0.01[33]. The Mahalanobis distance was then computed between all the possible 56-member couples of years at the grid-cell scale based on the following equation:

$$MD_{ij} = \sqrt{\sum_{k=1}^{N} \frac{\left(x_{jk} - x_{ik}\right)^2}{\sigma_k^2}} \quad (6)$$

where $j$ and $i$ belong to the first and second sub-period, respectively; $x_{jm}$ and $x_{im}$ are the values of the principal component $k$ in the year $j$ and $i$, $N$ represents the number of retained principal components, and $\sigma_k^2$ refers to the standard deviation of the principal component $k$. Low MD scores represent similar climate conditions between the two sub-periods, high MD scores the opposite. For each grid-cell, the minimum Mahalanobis distance ($MD_{min}$) was then selected, and its corresponding couple of years were identified as potential CA years.

We assessed the statistical significance of the obtained $MD_{min}$. Considering that the chi distribution provides a null distribution for (non-squared) Mahalanobis distances, the obtained $MD_{min}$ can be expressed probabilistically as percentiles of a *chi* distribution with degrees of freedom corresponding to the number of dimensions in which $MD_{min}$ was measured (i.e., $N$ in Eq. (6)). Following Mahony et al.[33], we considered the 95th percentile of the associated *chi* distribution to identify the upper threshold of the representative analog. $MD_{min}$ whose corresponding percentile is lower than the above-mentioned threshold, indicates a statistically similar climate between those two years (i.e., CA years).

Climate analogs were found not significant for a minority of grid cells (4.2%), and these areas were therefore excluded from the following analyses. For the remaining grid cells, we estimated the direct physiological effect of $eCO_2$ on GPP as follows:

$$eCO_2(dir)_{obs} = \frac{\Delta GPP_{obs}^{CA}}{\Delta CO_2^{CA}} \quad (7)$$

where $\Delta GPP_{obs}^{CA}$ is the change in growing-season $GPP_{obs}$ computed between CA years, and $\Delta CO_2^{CA}$ is the corresponding variation in atmospheric $CO_2$ concentration acquired from the Earth System Research Laboratory of NOAA[84] (https://www.esrl.noaa.gov/gmd/ccgg/trends/). The analysis was complemented by the estimation of the combined direct and indirect effect of $eCO_2$ on GPP quantified as:

$$eCO_2(net)_{obs} = \frac{\Delta GPP_{obs}}{\Delta CO_2} \quad (8)$$

where $\Delta GPP_{obs}$ and $\Delta CO_2$ represent the change in the mean growing-season $GPP_{obs}$ and $CO_2$ concentration, respectively, computed between the two sub-periods. We finally derived the observed indirect effect of $eCO_2$ on vegetation photosynthesis via associated climate change in a given period by combining Eq. (7) and Eq. (8) as follows:

$$eCO_2(ind)_{obs} = \frac{\Delta GPP_{obs}}{\Delta CO_2} - \frac{\Delta GPP_{obs}^{CA}}{\Delta CO_2^{CA}} \quad (9)$$

The approach assumes that both direct and indirect $eCO_2$ effects on GPP are annual and do not have legacy effects, and that $eCO_2$ is the dominant factor of climate change in the short-term. The above-described analyses were carried out at the grid-cell scale and separately for the periods 1982–1996 and 2000–2014. Changes between the two periods were then used to quantify the historical variations in the indirect effect of $eCO_2$ from observations (Fig. 1e, f).

To further test the robustness of our methods, a set of additional experiments were produced. First, the observed direct effect of $eCO_2$ was estimated using an alternative approach based on multiple non-linear regression model (i.e., Eq. (4)) in combination with CRU v4.05 climate dataset and $GPP_{obs}$. The obtained results, expressed as $eCO_2(dir)_{obs-RM}$, were confronted with climate analog-derived $eCO_2(dir)_{obs}$ estimates (i.e., obs-RM and obs in Fig. 3a). Second, the climate analog approach presented above and applied to observations was also implemented with CMIP6 model outputs in fully-coupled experiments to verify the consistency with results obtained from factorial simulations described in Eq. (3) (Supplementary Fig. 28).

## Statistical analysis

To explore the dynamics of the indirect effect of $eCO_2$ on vegetation carbon uptake during the historical period, we quantified the temporal changes in model-based $eCO_2(ind)$ and observation-based $eCO_2(ind)_{obs}$ retrieved for the two independent periods 1982–1996 and 2000–2014. The significance of the emerging changes was assessed through $t$ test. Results presented in the main text refer to analyses conducted over 15-year time windows. Results obtained for different temporal window lengths (12 and 16 years) are quantified as well to verify the robustness of our results (Fig. 1a and Supplementary Fig. 1).

In exploring the projected changes in $eCO_2(ind)$ from CMIP6 model simulations, we refer to six 15-year consecutive and independent periods, namely 2011–2025, 2026–2040, 2041–2055, 2056–2070, 2071–2085, 2086–2100 (Fig. 2a). We considered a series of not overlapped temporal windows to eliminate the possible impact of autocorrelation. To better disentangle the signal of future variation in $eCO_2(ind)$, we compared the $eCO_2(ind)$ during the last 15 years (2086–2100) against that one originating from the first 15 years of the historical period investigated here (1982–1996) and assessed the statistical significance of the change through $t$ test.

Furthermore, to properly represent the generality of the relationships between the direction and extent of changes in $eCO_2(ind)$ and local aridity conditions, we performed binned average analysis across environmental gradients. Such spatial averaging minimizes the uncertainty originating from spatial heterogeneity (e.g., the difference in topography and vegetation type) that may randomly affect the control of climate change on vegetation carbon uptake (Figs. 1d and 4c, d).

## Data availability

All datasets used in this study are publicly available as referenced in Methods. Source data are provided with this paper.

## Code availability

The custom MATLAB (R2023a) codes written to read and analyze data and generated figures are publicly available at https://doi.org/10.5281/zenodo.10451254.

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

## Acknowledgements
W.W. was supported by the National Natural Science Foundation of China (grant no. U2240218). G.F. was supported by the Horizon Europe Project ECO2ADAPT (grant no. 101059498).

## Author contributions
W.W. and Z.C. conceived and designed the research; Z.C. implemented the data analysis; G.F. and A.C. contributed analysis ideas; Z.C. interpreted the results and drafted the initial manuscript; W.W., G.F., and A.C. provided suggestions and further improved writing. All authors approved the final version of this manuscript.

## Competing interests
The authors declare no competing interests.

## Additional information

**Peer review information** : *Nature Communications* thanks the anonymous reviewers for their contribution to the peer review of this work. A peer review file is available.

