## [Peer Review File · Nature Communications]

Transition from positive to negative indirect CO₂ effects on the vegetation carbon uptakeREVIEWER COMMENTS

Reviewer #1 (Remarks to the Author):

This paper assesses the impact of elevated atmospheric carbon dioxide concentrations on gross primary production. Using remote sensing and CMIP6 ESM output, the authors assess this response historically (1982-2014) as well as throughout the 21st century using SSP585. They also separate the effects into direct physiological effects (e.g., reduced stomatal opening) versus indirect radiative effects (e.g., warming) in both the ESMs (using both the normal and BGC runs) and observations. They conclude that increases in GPP in response to elevated atmospheric carbon dioxide concentrations are declining in recent decades and are projected to become negative during the 21st century, with implications for mitigation efforts.

I found the paper informative and easy to follow. I was confused why the authors didn't use the 1% runs which would have allowed them to directly estimate the direct and indirect effects of CO₂. It would also have allowed for a consistent methodology between both the historical and future periods. By using multiple methods to isolate the direct and indirect effects between ESMs in the historic and future periods, and then another method for isolating those impacts in the observational data, it begins to become difficult to compare the results. It might be worth performing the whole analysis using one method and comparing those results to what is presented here.

The paper also explores mechanisms behind the decline in the indirect CO₂ effect, but it seems to only investigate the aridity index, which may or may not be a reliable measure of dryness in the future (see Berg, et al. 2022: <https://agupubs.onlinelibrary.wiley.com/doi/full/10.1029/2022EF003259>). As Berg, et al. points out, there is no need to use an index in future simulations since the ESMs simulate the entire hydrologic cycle. Also, were other mechanisms investigated?

Minor comments:

I would recommend using the median across the ESM ensemble rather than the mean to reduce the impact of outliers.

L27 I would add how far into the future it is expected to become negative since this is investigated.

L84-85 change 'to global carbon cycle' to 'to the global carbon cycle'

L91-93 I would change the last sentence here to not call these previous studies 'impracticable', but to instead restructure to say how the present study accounts for these covarying factors and therefore progresses the science from these previous studies.

L124 comma should be replaced with period.

L125- Stated that there are hotspots in northern Europe and Siberia, but it appears that there are a lot of

blue regions there as well.

L139 should also reference Figure 1d

Figure 3b caption is unclear—what histograms?

Figure 2- what is the uncertainty of the 1982-1996 horizontal line? It's shown for all the other periods so it would be good to show that here as well.

Figure S8 would be easier to read with a more varied colorbar.

L669- Should soil moisture be used here? I would think that would have more of a direct control on GPP than precipitation, since precipitation will not account for the memory of the system. How well was this model able to explain the GPP data?

Reviewer #2 (Remarks to the Author):

Summary:

This manuscript investigates the impacts of elevated atmospheric CO₂ concentration on vegetation carbon uptake using data (satellite observations) and Earth system models. The authors found that while there was an initial increase in GPP from eCO₂, they expect this positive effect to turn negative later on in this century – especially in high, northern latitudes. This paper was well written and the analyses were thoroughly conducted and described. I have a few specific comments below that will help clarify parts of the manuscript.

Specific comments

- In abstract, could you please add approximately when the model shows the eCO₂ induced change is expected to turn negative? Maybe an average of when the 5 models show that happening – a broad description would be fine (mid-century, end of century, etc.).
- Could you provide a table with model scenario descriptions and ensemble of models used in the main text? I see that the models used are in the SI – as a reader I would prefer that to be in the main text. It could be a truncated version of the detailed table, with the detailed table still being available in the SI.
- The authors did a nice job addressing the limitations of ESMs not representing disturbances well, and I appreciate the paragraph explaining how this may be a limitation of their results (lines 325-344). The authors explain how observation-based estimates are lower than model simulations in the historical period. Could the authors add a sentence here hypothesizing how that may impact their future simulation results (i.e., do they expect their results would show a greater decline if disturbances and biotic agents were better represented in models?)

First of all, we would like to thank the two reviewers for their insightful and constructive comments. In the revision of the manuscript, we tried to address all their comments and suggestions to improve the robustness of the analyses and the clarity of the interpretation.

In the following, we respond to each reviewer's comment by referring to line numbers of the revised tracked version, when not differently indicated. References cited along these responses are reported at the bottom of the document.

Reviewer #1: Questions and our responses

This paper assesses the impact of elevated atmospheric carbon dioxide concentrations on gross primary production. Using remote sensing and CMIP6 ESM output, the authors assess this response historically (1982-2014) as well as throughout the 21st century using SSP585. They also separate the effects into direct physiological effects (e.g., reduced stomatal opening) versus indirect radiative effects (e.g., warming) in both the ESMs (using both the normal and BGC runs) and observations. They conclude that increases in GPP in response to elevated atmospheric carbon dioxide concentrations are declining in recent decades and are projected to become negative during the 21st century, with implications for mitigation efforts.

I found the paper informative and easy to follow.

We thank the reviewer for her/his overall positive comments.

Major comments:

1. I was confused why the authors didn't use the 1% runs which would have allowed them to directly estimate the direct and indirect effects of CO₂. It would also have allowed for a consistent methodology between both the historical and future periods. By using multiple methods to isolate the direct and indirect effects between ESMs in the historic and future periods, and then another method for isolating those impacts in the observational data, it begins to become difficult to compare the results. It might be worth performing the whole analysis using one method and comparing those results to what is presented here.

Response: We agreed that the idealized 1% per year increasing CO₂ experiment provides simulations with fully-, biogeochemically-, and radiatively-coupled modes ("1pctCO₂", "1pctCO₂-bgc" and "1pctCO₂-rad" in the CMIP6 terminology, respectively). The availability of these three factorial experiments indeed allows the robust estimate of the direct and indirect effects of eCO₂ on the global carbon uptake. However, we argued

that the increasing rate of CO₂ in “1pctCO₂”, “1pctCO₂-bgc” and “1pctCO₂-rad” experiments is substantially higher over the historical period compared to that recorded with observations (Supplementary Figure 9, and also Figure 1 in Jones et al. (2016)). Such difference limits the comparability between model results obtained under these idealized experiments and results retrieved from observations.

To maximize confidence in our results, we still considered crucial to focus on CMIP6 model outputs generated under CO₂ conditions during the historical period that are consistent with observations. Furthermore, we argued that compared with results based on idealized simulations (i.e., “1pctCO₂”, “1pctCO₂-bgc” and “1pctCO₂-rad”), our approach provides a more intuitive explanation of the specific time when the initial positive effect of eCO₂-induced climate change on global vegetation carbon uptake will turn negative (Figure 2a), and when the negative indirect CO₂ effect will overcome the positive direct CO₂ effect (Supplementary Figure 8) under no climate policies. In this respect, our approach allows us to communicate key messages to policymakers and relevant communities ultimately fostering the development of effective climate adaption and mitigation strategies.

However, we recognized the value of the idealized CMIP6 simulations to derive an unequivocal signal of the direct physiological effect of eCO₂ as stressed by the reviewer. Therefore, we tested the robustness of our findings against the “factorial method“ by running a series of additional experiments based on CMIP6 model simulations under the idealized 1%yr⁻¹ increasing CO₂ experiments. We estimated the indirect effect of eCO₂ on growing-season GPP under the 1%yr⁻¹ increasing CO₂ experiments (hereafter eCO₂(ind)_{1%}) by using Eq. (3) in combination with simulated GPP in “1pctCO₂” and “1pctCO₂-bgc” experiments and the prescribed increasing rate of CO₂. Meanwhile, we estimated the direct CO₂ effect under the 1%yr⁻¹ increasing CO₂ experiments (hereafter eCO₂(dir)_{1%}) similarly based on the following equation:

$$eCO_2(dir)_{1\%} = \frac{\delta GPP^{FULL}_{1\%} - \delta GPP^{RAD}_{1\%}}{\delta CO_2(1\%)} \quad (14)$$

where $\delta GPP^{FULL}_{1\%}$ and $\delta GPP^{RAD}_{1\%}$ are the trends in growing-season GPP in the fully-coupled experiment (i.e., “1pctCO₂”) and the radiatively-coupled experiment (i.e., “1pctCO₂-rad”), respectively; $\delta CO_2(1\%)$ represents the trend in atmospheric CO₂ concentration in these idealized experiments (blue line in Supplementary Figure 9). To explore the temporal dynamic of indirect and direct effects of eCO₂ on vegetation carbon uptake with the 1%yr⁻¹ increasing rate of atmospheric CO₂ concentration, we calculated the changes in eCO₂(ind)_{1%} and eCO₂(dir)_{1%} between the Year 14-28 (period from the 14th year to 28th year for the entire 140 years) and the Year 120-134. These two periods were selected because the mean annual atmospheric CO₂ concentration during Year 14-28 and Year 120-134 in “1pctCO₂” is similar to that one during 1982-1996 and 2086-2100 in historical and future simulations.

An ensemble of six ESMs shows that global $e\text{CO}_2(\text{ind})_{1\%}$ decreases significantly by $0.33 \text{ gC m}^{-2} \text{ ppm}^{-1}$ between the Year 14-28 and the Year 120-134 ($p < 0.01$, t -test) (Supplementary Figure 10a,b). Such decreasing signal is statistically significant ($p < 0.05$) over 48.2% of global vegetated land and prominently in northern high-latitudes, eastern Australia and central Africa (Supplementary Figure 10c,d). The global mean magnitude and the spatial patterns of simulated changes in $e\text{CO}_2(\text{ind})_{1\%}$ between the Year 14-28 and the Year 120-134 are largely consistent with those retrieved from the historical and future simulations (Figure 2a-c). The discrepancy between these two sets of simulations occurs in the tropics and particularly in central Africa, suggesting possible divergent variations in tropical carbon-climate feedback under different growth rates of atmospheric CO_2 concentration in these regions. Furthermore, results based on multi-model ensemble mean show that global $e\text{CO}_2(\text{dir})_{1\%}$ decreases from $1.54 \pm 0.27 \text{ gC m}^{-2} \text{ ppm}^{-1}$ in Year 14-28 to $0.46 \pm 0.07 \text{ gC m}^{-2} \text{ ppm}^{-1}$ in Year 120-134 (Supplementary Figure 11a,b). Spatially, 76.0% of the global vegetated land exhibits a significant ($p < 0.05$) decrease in $e\text{CO}_2(\text{dir})_{1\%}$ between the Year 14-28 and the Year 120-134 (Supplementary Figure 11c). Combining these concurrent temporal changes (Supplementary Figure 10c), we found that 69.4% of global vegetated land could experience the same direction of change in $e\text{CO}_2(\text{ind})_{1\%}$ and $e\text{CO}_2(\text{dir})_{1\%}$ (i.e., “+ +” and “- -” in Supplementary Figure 11d) between the Year 14-28 and the Year 120-134, while the remaining 30.6% could manifest opposite directions of change (i.e., “+ -” and “- +”). The concurrent decrease in $e\text{CO}_2(\text{ind})_{1\%}$ and $e\text{CO}_2(\text{dir})_{1\%}$ (“- -”) is the most pervasive case and occurs in 53.2% of global vegetated land, which is consistent with results based on historical and future scenario simulations (Figure 3).

Furthermore, we exploited the “1pctCO2-rad” experiment to further assess the validity of our multiple non-linear regression framework (i.e., Eq. (4)) employed to estimate the direct effect of $e\text{CO}_2$ on vegetation carbon uptake. To this aim, we applied the multiple non-linear regression framework (i.e., Eq. (4)) to simulations run with the fully-coupled mode (i.e., “1pctCO2” experiment), and compared the associated estimates (hereafter $e\text{CO}_2(\text{dir})_{1\%-\text{RM}}$) against analogous estimates obtained from the factorial experiment described above (i.e., $e\text{CO}_2(\text{dir})_{1\%}$ calculated based on Eq. (14)). At the global scale, $e\text{CO}_2(\text{dir})_{1\%-\text{RM}}$ estimated by multi-model ensemble mean decreases from $1.65 \pm 0.31 \text{ gC m}^{-2} \text{ ppm}^{-1}$ in Year 14-28 to $0.53 \pm 0.13 \text{ gC m}^{-2} \text{ ppm}^{-1}$ in Year 120-134 (Supplementary Figure 12a,b). The decreasing magnitude of $e\text{CO}_2(\text{dir})_{1\%-\text{RM}}$ ($-1.12 \text{ gC m}^{-2} \text{ ppm}^{-1}$, or -68.0%) is similar to that one obtained from $e\text{CO}_2(\text{dir})_{1\%}$ ($-1.08 \text{ gC m}^{-2} \text{ ppm}^{-1}$, or -70.0%) (Supplementary Figure 11a,b), suggesting the high consistency between the two sets of results in terms of global mean level. The global patterns of the direct CO_2 effect in Year 14-28 and in Year 120-134 simulated by the use of the non-linear regression model (i.e., $e\text{CO}_2(\text{dir})_{1\%-\text{RM}}$) are also generally consistent with those obtained directly from factorial experiments (i.e., $e\text{CO}_2(\text{dir})_{1\%}$), as confirmed by the high significant ($p < 0.01$) spatial correlation coefficient (0.78 and 0.64, respectively) computed over the vegetated grid-cells

(Supplementary Figure 12c-f). Moreover, results also show that change in direct CO₂ effect between Year 14-28 and Year 120-134 estimated by the non-linear regression model strongly agrees with that directly derived from factorial experiments, with correlation coefficient reaching 0.72 ($p < 0.01$) (Supplementary Figures 11c and 12g,h).

The high consistency of results obtained from non-linear regression and directly from factorial experiments, as described above, proves the suitability of our methods to investigate the direct effect of eCO₂ on global vegetation carbon uptake. In addition, we compared results based on the climate analog approach (Eq. (7)) against analogous estimates based on multiple non-linear regression (Eq. (4)) and factorial experiments (Eq. (14)) under the 1% per year increasing CO₂ experiment (Supplementary Figure 13). As evident, the three sets of results show good agreement, demonstrating the consistency and comparability of results obtained from three different methods.

Action taken: We have described the aforementioned experiments in the new Supplementary Text 4 (“Results from the idealized 1%yr⁻¹ increasing CO₂ experiments” section) and briefly recalled in the main text (see lines 230-239) with references to the new Supplementary Figures 9-13. We believe that these new analyses help (1) to clarify the rationale of using historical and future scenario simulations instead of simulations under the idealized 1% per year increasing CO₂ experiment, (2) to further improve the robustness of our finding about the concurrent decline in both indirect and direct CO₂ effects on vegetation carbon uptake, and (3) to further support the validity of the use of multiple methods (i.e., factorial experiment, non-linear regression model, and climate analog approach).

Figure S9. Interannual changes in atmospheric CO₂ concentration at the global scale in idealized 1%yr⁻¹ increasing CO₂ simulations (i.e., “1pctCO₂”) and in historical and future scenario (SSP5-8.5) simulations, respectively.

Figure S10. (a) Mean indirect effect of eCO_2 on growing-season GPP via associated climate change ($eCO_2(\text{ind})_{1\%}$) during Year 14-28 and Year 120-134 in idealized $1\%yr^{-1}$ increasing CO_2 experiments, as derived from an ensemble of six ESMs (details in Supplementary Text 4). Error bars represent the standard error of effects derived from ensemble members. Δ expresses the mean of difference in $eCO_2(\text{ind})_{1\%}$ between the two periods. Statistical significance of the difference is assessed by t -test. (b) Frequency distribution of $eCO_2(\text{ind})_{1\%}$ at the global scale during Year 14-28 and Year 120-134, as estimated by an ensemble of six ESMs. Distribution averages are shown as dotted horizontal lines. (c) Spatial pattern of difference in $eCO_2(\text{ind})_{1\%}$ between the two periods (Year 120-134 versus Year 14-28) derived from an ensemble of six ESMs. Non-vegetated areas are excluded in our analysis and are shown in grey. Regions labelled by black dots indicate differences that are statistically significant (t -test, $p < 0.05$). Dots are spaced 3° in both latitude and longitude, and statistics were computed over $9^{\circ} \times 9^{\circ}$ spatial moving windows. (d) Zonal medians of difference in $eCO_2(\text{ind})_{1\%}$ between the two periods (Year 120-134 versus Year 14-28) simulated by an ensemble of six ESMs at 5° latitudinal resolution. Corresponding interquartile ranges of model simulation are shown as shaded band.

Figure S11. (a) Mean direct (physiological) effect of $e\text{CO}_2$ on growing-season GPP ($e\text{CO}_2(\text{dir})_{1\%}$) during Year 14-28 and Year 120-134 in idealized $1\% \text{yr}^{-1}$ increasing CO_2 experiments, as derived from an ensemble of six ESMs (details in Supplementary Text 4). Error bars represent the standard error of effects derived from ensemble members. Δ expresses the mean of difference in $e\text{CO}_2(\text{dir})_{1\%}$ between the two periods. Statistical significance of the difference is assessed by t -test. (b) Frequency distribution of $e\text{CO}_2(\text{dir})_{1\%}$ at the global scale during Year 14-28 and Year 120-134, as estimated by an ensemble of six ESMs. Distribution averages are shown as dotted horizontal lines. (c) Spatial pattern of difference in $e\text{CO}_2(\text{dir})_{1\%}$ between the two periods (Year 120-134 versus Year 14-28) derived from an ensemble of six ESMs. Non-vegetated areas are excluded in our analysis and are shown in grey. Regions labelled by black dots indicate differences that are statistically significant (t -test, $p < 0.05$). Dots are spaced 3° in both latitude and longitude, and statistics were computed over $9^\circ \times 9^\circ$ spatial moving windows. (d) Spatial pattern of relationship between changes in indirect effect of $e\text{CO}_2$ on growing-season GPP via associated climate change ($e\text{CO}_2(\text{ind})_{1\%}$) and $e\text{CO}_2(\text{dir})_{1\%}$ between the two periods (Year 120-134 versus Year 14-28), where “--” represents decrease in $e\text{CO}_2(\text{ind})_{1\%}$ and decrease in $e\text{CO}_2(\text{dir})_{1\%}$, “-+” represents decrease in $e\text{CO}_2(\text{ind})_{1\%}$ and increase in $e\text{CO}_2(\text{dir})_{1\%}$ and so on. Legend shows the fraction of vegetated areas for each thematic class (i.e., “--”, “-+”, “+-” and “++”).

Figure S12. (a) Mean direct (physiological) effect of $e\text{CO}_2$ on growing-season GPP ($e\text{CO}_2(\text{dir})_{1\%-\text{RM}}$) during Year 14-28 and Year 120-134, as estimated by multi-model ensemble simulations in idealized $1\% \text{yr}^{-1}$ increasing CO_2 experiments in combination with the non-linear regression model (i.e., Eq. (4)) (details in Supplementary Text 4). Error bars represent the standard error of effects derived from ensemble members. Δ expresses the mean of difference in $e\text{CO}_2(\text{dir})_{1\%-\text{RM}}$ between the two periods. (b) Frequency distribution of $e\text{CO}_2(\text{dir})_{1\%-\text{RM}}$ at the global scale during Year 14-28 and Year 120-134. Distribution averages are shown as dotted horizontal lines. (c) Spatial pattern of $e\text{CO}_2(\text{dir})_{1\%-\text{RM}}$ during Year 14-28 derived from multi-model ensemble simulations in combination with the non-linear regression model. Non-vegetated areas are excluded in our analysis and are shown in grey. (d) Comparison of $e\text{CO}_2(\text{dir})_{1\%-\text{RM}}$ during Year 14-28 against analogous estimates directly by factorial experiments (i.e., Eq. (14)). Each symbol represents one vegetated grid-cell. Red dotted lines indicate the best-fit

with equation provided on each subplot. (e-h) Same as (c and d), but for $e\text{CO}_2(\text{dir})_{1\%-\text{RM}}$ during Year 120-134 and for the difference between the two periods (Year 120-134 versus Year 14-28), respectively.

Figure S13. Zonal medians of direct (physiological) effect of $e\text{CO}_2$ on growing-season GPP (a) during Year 14-28 and (b) Year 120-134 and (c) their difference (Year 120-134 versus Year 14-28) at 5° latitudinal resolution, as estimated by factorial experiment (Eq. (14)), by non-linear regression model (Eq. (4)) and by climate analog approach (Eq. (7)), respectively. Corresponding interquartile ranges of multi-model simulation are shown as shaded band.

2. The paper also explores mechanisms behind the decline in the indirect CO_2 effect, but it seems to only investigate the aridity index, which may or may not be a reliable measure of dryness in the future (see Berg, et al. 2022: <https://agupubs.onlinelibrary.wiley.com/doi/full/10.1029/2022EF003259>). As Berg, et al. points out, there is no need to use an index in future simulations since the ESMs simulate the entire hydrologic cycle. Also, were other mechanisms investigated?

Response: We agreed with the reviewer that hydrological variables (e.g., soil moisture) directly simulated by ESMs could better represent land surface aridity in the future compared to dryness indices such as Aridity Index and Palmer Drought Severity Index (PDSI). Therefore, we have performed additional analyses based on surface soil moisture (SM_{surf} , 0-10cm) and total soil moisture (SM_{total} , depth depending on models, see Table 1) simulated directly by ESMs to better capture the hydrological constraints and further investigate the mechanisms behind the decline in indirect CO_2 effect.

To this aim, we downloaded simulated SM_{surf} and SM_{total} during the historical and future scenario periods. Considering that the hydrologically active soil depth varies greatly among the models (from 2 m in UKESM1-0-LL to 35.18 m in E3SM-1-1), SM_{total} as well as SM_{surf} was converted from the original gravimetric unit (kg m^{-2}) to volumetric unit ($\text{m}^3 \text{m}^{-3}$) by dividing the gravimetric soil water content by the corresponding soil depth. Such conversion allows for the comparison of results obtained from different models and the development of more robust multi-model ensembles of soil moisture. Variations in SM_{total} and SM_{surf} were expressed in terms

of relative changes (%) with respect to their average values computed for the baseline period (e.g., 1982-1996) (Berg et al., 2017).

We first exploited the CMIP6 simulations to quantify the spatiotemporal variations in aridity conditions, expressed in terms of surface (0-10cm) soil moisture (SM_{surf}). Results indicate a projected widespread decline in terrestrial water availability by the end of the century compared to the current conditions (82.6% of global vegetated land exposed to a reduction in SM_{surf} , Figure 4a). At the global level and based on multi-model means (i.e., CMIP6_{SMA}), we estimated a significant decrease in SM_{surf} during 2086-2100 by 7.3% ($p < 0.01$, t -test) compared to analogous estimates obtained for the 1982-1996 period (Figure 4b). Similar drying patterns emerge for individual model runs (Supplementary Figure 15), for total soil moisture (SM_{total}), for a widely-used aridity index (defined as the ratio of annual precipitation to potential evapotranspiration, P/PET) (Supplementary Figures 16 and 17), and are further corroborated by previous studies focusing on dryness indices (Dai, 2013) and hydrological regimes (Berg et al., 2017; Cook et al., 2020).

To investigate the relationship between change in $eCO_2(ind)$ and land surface drying/wetting, we then averaged the change in $eCO_2(ind)$ across gradients of mean annual SM_{surf} during 1982-1996 and the corresponding change in SM_{surf} (i.e., 2000-2014 versus 1982-1996, and 2086-2100 versus 1982-1996). $SM_{surf} = 0.26 \text{ m}^3 \text{ m}^{-3}$ generally corresponds to $P/PET = 1$ at the mean annual scale (Supplementary Figure 17b), which is widely treated as the threshold between non-humid and humid regions (UNEP, 1997; Ukkola et al., 2016). We found that $eCO_2(ind)$ generally declines (enhances) with the land drying (wetting) in humid regions ($SM_{surf} > 0.26 \text{ m}^3 \text{ m}^{-3}$, Supplementary Figure 4a) in both historical and scenario simulations (Figure 4c,d). However, in water-limited environments ($SM_{surf} < 0.26 \text{ m}^3 \text{ m}^{-3}$), the weakened negative $eCO_2(ind)$ along with the land drying results in a negative relationship between changes in $eCO_2(ind)$ and SM_{surf} (Figure 4c,d and Supplementary Figure 7). CO_2 and drought-related enhancement in growing-season water-use efficiency (WUE) (Supplementary Figure 18), relax the water limitation to vegetation growth, especially over semi-arid climate zones, and may consequently limit the negative trend in $eCO_2(ind)$ (Figure 4d). In addition, for water-limited environments, a decrease in $eCO_2(ind)$ occurs consistently under both land drying and wetting, indicating the possible importance of other factors, such as vegetation type and species diversity, in modulating the vegetation response to climate change.

We repeated the analyses using different temporal window lengths (12 and 16 years), and adopting an alternative soil moisture variable (i.e., SM_{total}) (Supplementary Figures 16 and 19). The high consistency emerging among multiple sets of results demonstrates the substantial independence of our results on the proxy of terrestrial water availability and the selection of time-window length.

Action taken: We have added the aforementioned results based on SM_{surf} under 15-year moving windows

in the revised version of the manuscript (see lines 247-280). Sensitivity analysis for different temporal window lengths (12 and 16 years) has been reported in Supplementary Information, and the corresponding Figure 4 and Supplementary Figures 4, 15, and 19 also have been added in the revised version. To keep the main text reasonably concise, results based on SM_{total} and original results based on aridity index were shown in Supplementary Information (Supplementary Figures 16 and 17), and briefly recalled in the main text (see lines 281-285). The methodological details related to SM_{surf} and SM_{total} provided by CMIP6 ESMs and their pre-processing were described in the Materials and Methods section (see lines 549-559).

Figure 4 | Sensitivity of indirect CO_2 effect on terrestrial water availability. (a) Spatial pattern of relative change in surface soil moisture (SM_{surf}) between the historical and future periods (2086-2100 versus 1982-1996) derived from CMIP6_{SMA}. Regions labelled by black dots indicate changes that are statistically significant (t -test, $p < 0.05$). Dots are spaced 3° in both latitude and longitude, and statistics were computed over $9^\circ \times 9^\circ$ spatial moving windows. (b) Global mean SM_{surf} derived from CMIP6_{SMA} during the period 1982-1996, 2000-2014, and 2086-2100, respectively. Numbers refer to change in SM_{surf} relative to 1982-2016. (c) Difference in indirect effect of eCO_2 on growing-season GPP via associated climate change ($eCO_2(ind)$) between the periods 1982-1996 and 2000-2014 derived from CMIP6_{SMA}, binned as a function of corresponding changes in SM_{surf} and mean annual SM_{surf} (Supplementary Figure 4a). $SM_{surf} = 0.26 m^3 m^{-3}$ (i.e., the green solid line) overall corresponds to $P/PET = 1$ at the mean annual scale, that is the threshold between non-humid and humid regions (Supplementary Figure 17b). Black dots indicate bins with differences that are statistically significant (t -test, $p < 0.05$). (d) Same as (c), but for difference between the periods 1982-1996 and 2086-2100.

Figure S4. (a and b) Spatial pattern of surface soil moisture (SM_{surf}) and total soil moisture (SM_{total}) during the period 1982-1996 derived from CMIP6_{SMA}, respectively. (c and d) Spatial pattern of relative changes in SM_{surf} and SM_{total} between the periods 1982-1996 and 2000-2014 derived from CMIP6_{SMA}, respectively. Regions labelled by black dots indicate changes that are statistically significant (t -test, $p < 0.05$). Dots are spaced 3° in both latitude and longitude, and statistics were computed over $9^\circ \times 9^\circ$ spatial moving windows.

Figure S15. (a-g) Spatial pattern of relative change in surface soil moisture (SM_{surf}) between the historical and future periods (2086-2100 versus 1982-1996) derived from seven CMIP6 ESMs, respectively. Regions labelled by black dots indicate changes that are statistically significant (t -test, $p < 0.05$). Dots are spaced 3° in both latitude and longitude, and statistics were computed over $9^\circ \times 9^\circ$ spatial moving windows. (h) Global mean change in SM_{surf} between the historical and future periods derived from seven CMIP6 ESMs, respectively. Two asterisks indicate that the change is statistically significant (t -test, $p < 0.05$).

Figure S16. (a) Spatial pattern of relative change in total soil moisture (SM_{total}) between the historical and future periods (2086-2100 versus 1982-1996) derived from CMIP6_{SMA}. Regions labelled by black dots indicate changes that are statistically significant (t -test, $p < 0.05$). Dots are spaced 3° in both latitude and longitude, and statistics were computed over $9^\circ \times 9^\circ$ spatial moving windows. (b) Global mean SM_{total} derived from CMIP6_{SMA} during the period 1982-1996, 2000-2014, and 2086-2100, respectively. Numbers refer to change in SM_{total} relative to 1982-2016. (c) Difference in indirect effect of eCO_2 on growing-season GPP via associated climate change ($eCO_2(ind)$) between the periods 1982-1996 and 2000-2014 derived from CMIP6_{SMA}, binned as a function of corresponding changes in SM_{total} and mean annual SM_{total} (Supplementary Figure 4b). $SM_{total} = 0.23 m^3 m^{-3}$ (i.e., the green solid line) overall corresponds to $P/PET = 1$ at the mean annual scale, that is the threshold between non-humid and humid regions (Supplementary Figure 17b). Black dots indicate bins with differences that are statistically significant (t -test, $p < 0.05$). (d) Same as (c), but for difference between the periods 1982-1996 and 2086-2100.

Figure S17. (a) Spatial pattern of the ratio of mean annual precipitation to potential evapotranspiration (P/PET) during the period 1982-1996 derived from CMIP6_{SMA}. (b) Relationship between P/PET and surface soil moisture (SM_{surf} , in red) and total soil moisture (SM_{total} , in blue) during 1982-1996 derived from CMIP6_{SMA}. Each symbol represents one vegetated grid-cell. Solid lines indicate the best-fit with equations provided. According to fit equations, $\text{P/PET}=1$, i.e., the threshold between non-humid and humid regions generally corresponds to $\text{SM}_{\text{surf}} \approx 0.26 \text{ m}^3 \text{ m}^{-3}$ and $\text{SM}_{\text{total}} \approx 0.23 \text{ m}^3 \text{ m}^{-3}$ at the mean annual scale. (c) Spatial pattern of difference in P/PET between the historical and future periods (2086-2100 versus 1982-1996) derived from CMIP6_{SMA}. Regions labelled by black dots indicate differences that are statistically significant (t -test, $p<0.05$). Dots are spaced 3° in both latitude and longitude, and statistics were computed over $9^\circ \times 9^\circ$ spatial moving windows. (d) Global mean P/PET derived from CMIP6_{SMA} during the period 1982-1996, 2000-2014, and 2086-2100, respectively. Numbers refer to change in P/PET relative to 1982-2016. (e) Difference in indirect effect of eCO_2 on growing-season GPP via associated climate change ($\text{eCO}_2(\text{ind})$) between the periods 1982-1996 and 2000-2014 derived from CMIP6_{SMA}, binned as a function of corresponding changes in P/PET and mean annual P/PET in (a). Black dots indicate bins with differences that are statistically significant (t -test, $p<0.05$). (f) Same as (e), but for difference between the periods 1982-1996 and 2086-2100.

Figure S19. (a) Difference in indirect effect of eCO_2 on growing-season GPP via associated climate change ($eCO_2(ind)$) between the periods 1982-1993 and 2000-2014 derived from CMIP6_{SMA}, binned as a function of corresponding changes in SM_{surf} and mean annual SM_{surf} (1982-1993). $SM_{surf}=0.26 \text{ m}^3 \text{ m}^{-3}$ (i.e., the green solid line) overall corresponds to $P/PET=1$ at the mean annual scale, that is the threshold between non-humid and humid regions (Supplementary Figure 17b). Black dots indicate bins with differences that are statistically significant (t -test, $p<0.05$). (b) Same as (a), but for difference between the periods 1982-1997 and 2085-2100.

Minor comments:

1. I would recommend using the median across the ESM ensemble rather than the mean to reduce the impact of outliers.

Response: In addition to simple model averaging, we also employed the Bayesian model averaging (BMA) method to integrate the multi-model ensemble (details in Supplementary Text 1). Two sets of results (based on BMA and simple model averaging) were compared to test if our model results were potentially affected by the different averaging methods. To retain this detailed test which could further improve the robustness of our findings, we had to use the mean across the ESM ensemble.

Furthermore, following the recommendation of the reviewer, we have expanded the analysis using alternative metrics to describe the ensemble results. We have calculated the changes in indirect CO_2 effect on global vegetation carbon uptake based on the multi-model ensemble median, and compared the obtained results against those based on the multi-model ensemble mean. Such ensemble median of historical simulations from seven CMIP6 ESMs shows that averaged across the global vegetated areas, the indirect CO_2 effect on vegetation carbon uptake has decreased by $0.14 \text{ gC m}^{-2} \text{ ppm}^{-1}$ between the periods 2000-2014 and 1982-1996, and is expected to decrease to $-0.10 \text{ gC m}^{-2} \text{ ppm}^{-1}$ during 2086-2100. As evident, results based on the multi-model ensemble median confirm that the initial positive effect of eCO_2 -induced climate change on

global vegetation carbon uptake has declined during recent decades and is expected to turn negative in the future, largely consistent with our original results based on the multi-model ensemble mean. All together, these signals further confirm the robustness of our results with respect to integration method of the multi-model ensemble.

2. L27 I would add how far into the future it is expected to become negative since this is investigated.

Response and **action taken**: eCO₂-induced climate change impact on global vegetation carbon uptake has turned negative in the early 21st century, and is expected to be firmly negative thereafter. We have added this temporal information into Abstract section (see line 27).

3. L84-85 change ‘to global carbon cycle’ to ‘to the global carbon cycle’

Response and **action taken**: We have changed ‘to global carbon cycle’ to ‘to the global carbon cycle’ (see line 78).

4. L91-93 I would change the last sentence here to not call these previous studies ‘impracticable’, but to instead restructure to say how the present study accounts for these covarying factors and therefore progresses the science from these previous studies.

Response and **action taken**: We have modified the expression of this sentence. Details are as follows: “Existing studies focusing on a single climate driver (e.g., temperature (Piao et al., 2014)), generally assumed that the effects are independent by neglecting the covariation and the interaction between drivers. Therefore, the assessment of the variations in total indirect effect of eCO₂ can only be partially represented.” (see lines 84-88).

5. L124 comma should be replaced with period.

Response and **action taken**: We have modified the text according to the reviewer’s suggestion (see line 118).

6. L125- Stated that there are hotspots in northern Europe and Siberia, but it appears that there are a lot of blue regions there as well.

Response and **action taken**: We have improved the description of the emerging hotspots in the revised version. To this aim, we have replaced the original “northern Europe and Siberia” with “Scandinavia and south-central Siberia” (see lines 120-121).

7. L139 should also reference Figure 1d

Response and **action taken**: We have also added Figure 1d as reference (see line 134).

8. Figure 3b caption is unclear—what histograms?

Response: In Figure 3b, diamond symbols and whiskers indicate the ensemble mean and standard error of modelled $eCO_2(\text{dir})$ during the seven independent periods, namely 1982-1996, 2011-2025, 2026-2040, 2041-2055, 2056-2070, 2071-2085, and 2086-2100. Bars in green and orange represent the relative contributions of $eCO_2(\text{ind})$ and $eCO_2(\text{dir})$ to net effect of eCO_2 ($eCO_2(\text{net})$) during the corresponding periods and derived from model ensemble mean.

Action taken: We have revised the caption of Figure 3b, to improve its clarity.

9. Figure 2- what is the uncertainty of the 1982-1996 horizontal line? It's shown for all the other periods so it would be good to show that here as well.

Response and **action taken**: According to the reviewer's suggestion, we have revised Figure 2 and added uncertainty of the 1982-1996 horizontal line as shaded band.

10. Figure S8 would be easier to read with a more varied colorbar.

Response and **action taken**: According to the reviewer's suggestion, we have revised Figure S8 with a more varied colorbar.

11. L669- Should soil moisture be used here? I would think that would have more of a direct control on GPP than precipitation, since precipitation will not account for the memory of the system. How well was this model able to explain the GPP data?

Response: We have performed a number of additional analyses in the revised version to test if replacing precipitation with soil moisture in the regression model could improve the model performance in explaining GPP dynamics.

More specifically, we replaced precipitation (P) in the multiple non-linear regression framework described in the main text (i.e., Eq. (4)) with surface soil moisture (SM_{surf}), and correspondingly, developed an additional regression framework as follows:

$$GPP = \beta(CO_2) + C_1(SM_{surf}) + C_2(VPD) + C_3(T_{min} \cdot VPD) + C_4(SM_{surf} \cdot CL) + C_5 + \varepsilon \quad (19)$$

We applied such new multiple non-linear regression (Eq. (19)) to observation-driven datasets, i.e. GPP_{obs} , CRU v4.05 climate dataset, and SM_{surf} provided by the Global Land Evaporation and Amsterdam Model (GLEAM v3.8a) (Martens et al., 2017), and compared against the original regression (Eq. (4)):

$$GPP = \beta(CO_2) + C_1(P) + C_2(VPD) + C_3(T_{min} \cdot VPD) + C_4(P \cdot CL) + C_5 + \varepsilon \quad (4)$$

According to the Akaike Information Criterion (AIC), the corrected Akaike Information Criterion (AICc) and the Bayesian Information Criterion (BIC), we found that Eq. (19) does not lead to a substantial improvement in model performance compared to Eq. (4). On the contrary, for a considerable part of global vegetated land, Eq. (19) has a lower model performance than Eq. (4) (Supplementary Figure 26a-c,g). Specifically speaking, global mean AIC, AICc, and BIC of Eq. (19) reach 160.06, 176.06, and 165.01 for the period 1982-1996, slightly lower than those of Eq. (4) (160.60, 176.60, and 165.55). Spatially, test results show that Eq. (19) has substantially lower AIC, AICc, and BIC (relative change < -5%) in only 5.4%, 4.2%, and 5.0% of global vegetated land than Eq. (4) for the period 1982-1996. By contrast, Eq. (19) has substantially higher AIC, AICc, and BIC (relative change > 5%) in 3.6%, 2.9%, and 3.3% of the globe than Eq. (4). The limited and less widespread decrease in AIC, AICc, and BIC of Eq. (19) compared to Eq. (4) suggests that replacing P with SM_{surf} would not result in a substantial model improvement in representing GPP dynamics. We further compared the observed direct effect of eCO_2 on GPP ($eCO_2(\text{dir})_{obs-RM}$) estimated by Eq. (19) against analogous estimates obtained from Eq. (4), to assess the potential impact of incorporating soil moisture information on regression results (Supplementary Figure 27a,b). Results show that $eCO_2(\text{dir})_{obs-RM}$ estimated by Eq. (19) exhibits a great consistency with that one estimated by Eq. (4) for the periods 1982-1996 and 2000-2014, with spatial correlation coefficient (r) reaching 0.92 ($p < 0.01$), and 0.86 ($p < 0.01$), respectively. The high consistency of results from Eq. (19) and Eq. (4) indicates that replacing P with SM_{surf} has a negligible impact on our results, and further demonstrates the robustness of our estimates of observed direct CO_2 effect.

We also applied Eq. (19) to CMIP6 model simulations under “historical” and “ssp585” experiments and compared associated results against those based on Eq. (4). In line with those based on observations as

mentioned above (Supplementary Figure 26a-c), results based on ensemble mean of seven ESMS also show that AIC, AICc, and BIC of Eq. (19) are substantially lower (relative change $< -5\%$) than those of Eq. (4) in only 7.6%, 6.7%, and 7.2% of global vegetated land for the period 1982-1996 (Supplementary Figure 26d-f,g). On the contrary, Eq. (19) has substantially higher AIC, AICc, and BIC (relative change $> 5\%$) in 4.6%, 3.9%, and 4.4% of the vegetated land compared with Eq. (4). Similar conclusions can also be derived in model performance assessments for various periods (e.g., 2000-2014, and 2086-2100). For synthesis purposes, only assessment results for the period 1982-1996 were shown here.

Furthermore, correlation analysis between multi-model means of simulated direct effect of eCO_2 on GPP ($eCO_2(\text{dir})$) based on Eq. (19) and that one based on Eq. (4) shows Pearson's correlation coefficient of 0.86 ($p < 0.01$), 0.85 ($p < 0.01$), and 0.85 ($p < 0.01$) for the periods 1982-1996, 2000-2014, and 2086-2100, respectively (Supplementary Figure 27c-e). Apparently, replacing P with SM_{surf} in the regression model has limited impact on our estimates of direct CO_2 effect, regardless of whether the regression model is driven by observation-based datasets or by model outputs.

To test the potential impact of replacing P with SM_{surf} on the inter-model spread, we compared the standard error computed over the seven sets of $eCO_2(\text{dir})$ estimates (corresponding to the seven ESMS) based on Eq. (19) against the analogous estimate based on Eq. (4). Results show that global mean standard error of estimates generated with SM_{surf} is always higher than that one generated with P across all the periods examined in this study (Supplementary Table 3). The larger inter-model spread deriving when replacing P with SM_{surf} in the regression model may be attributed to the difference in representations of the processes related to the soil-vegetation continuum, roots profile and water potentials across ESMS (Trugman et al., 2018; Li et al., 2022).

In light of the consideration reported above, in order to minimize inter-model spread and maximize model performance, we adopted P as indicator of water availability conditions in Eq. (4) in our study.

Action taken: In the revised version, we have added these contents into the new Supplementary Text 7 ("Incorporating soil moisture into regression framework" section) and briefly recalled them in the main text (see lines 680-690), and correspondingly, added the new Supplementary Figures 26 and 27, and Table 3.

Figure S26. (a-c) Performance comparison of Eq. (4) and Eq. (19) with observation-based datasets as inputs for the period 1982-1996 on the basis of the Akaike Information Criterion (AIC), the corrected Akaike Information Criterion (AICc) and the Bayesian Information Criterion (BIC). Relative change in AIC was calculated by $(AIC_{Eq.(19)} - AIC_{Eq.(4)})/AIC_{Eq.(4)}$. (d-f) Same as (a-c) but for Eq. (4) and Eq. (19) with multi-model simulations as inputs. (g) Fraction of vegetated land with different levels of relative change in AIC, AICc and BIC.

Figure S27. Comparisons of direct (physiological) effect of eCO_2 on growing-season GPP ($eCO_2(\text{dir})_{\text{obs-RM}}$) during the periods (a) 1982-1996 and (b) 2000-2014 estimated by Eq. (19) against that estimated by Eq. (4) with observation-based datasets as inputs. Each symbol represents one vegetated grid-cell. Red dotted lines indicate the best-fit with equation provided on each subplot. (c-e) Same as (a and b), but for estimates based on multi-model ensemble simulations (i.e., $eCO_2(\text{dir})$).

Table S3. Global mean direct physiological effect of eCO_2 on growing-season GPP (i.e., $eCO_2(\text{dir})$, in unit of $gC\ m^{-2}\ ppm^{-1}$) during the periods 1982-1996, 2000-2014, 1982-2014, and 2086-2100, as estimated by 7-member model ensemble simulations in combination with Eq. (4) and Eq. (19), respectively. Values are expressed as ensemble mean \pm standard error.

	1982-1996	2000-2014	1982-2014	2086-2100
Eq. (4)	1.93 \pm 0.28	1.49 \pm 0.12	1.83 \pm 0.24	0.65 \pm 0.12
Eq. (19)	2.05 \pm 0.29	1.47 \pm 0.14	1.87 \pm 0.27	0.61 \pm 0.15

References:

1. Berg, A., Sheffield, J. & Milly, P. Divergent surface and total soil moisture projections under global warming. *Geophys. Res. Lett.* **44**, 236–244 (2017).
2. Cook, B. et al. Twenty-first century drought projections in the CMIP6 forcing scenarios. *Earth's Future* **8**, e2019EF001461 (2020).
3. Dai, A. Increasing drought under global warming in observations and models. *Nat. Clim. Change* **3**, 52-58 (2013).
4. Jones, C. et al. C4MIP – The Coupled Climate–Carbon Cycle Model Intercomparison Project: experimental protocol for CMIP6. *Geosci. Model Dev.* **9**, 2853–2880 (2016).
5. Li, W. et al. Widespread increasing vegetation sensitivity to soil moisture. *Nat. Commun.* **13**, 3959 (2022).
6. Martens, B. et al. GLEAM v3: satellite-based land evaporation and root-zone soil moisture. *Geosci. Model Dev.* **10**, 1903-1925 (2017).
7. Piao, S. et al. Evidence for a weakening relationship between interannual temperature variability and northern vegetation activity. *Nat. Commun.* **5**, 5018 (2014).
8. Trugman, A., Medvigy, D., Mankin, J. & Anderegg, W. Soil moisture stress as a major driver of carbon cycle uncertainty. *Geophys. Res. Lett.* **45**, 6495–6503 (2018).
9. Ukkola, A. et al. Reduced streamflow in water-stressed climates consistent with CO₂ effects on vegetation. *Nat. Clim. Change* **6**, 75-78 (2016).
10. United Nations Environment Program (UNEP). World Atlas of Desertification. 182 (1997).

Reviewer #2: Questions and our responses

This manuscript investigates the impacts of elevated atmospheric CO₂ concentration on vegetation carbon uptake using data (satellite observations) and Earth system models. The authors found that while there was an initial increase in GPP from eCO₂, they expect this positive effect to turn negative later on in this century – especially in high, northern latitudes. This paper was well written and the analyses were thoroughly conducted and described. I have a few specific comments below that will help clarify parts of the manuscript.

We thank the reviewer for her/his overall positive comments.

Specific comments:

1. In abstract, could you please add approximately when the model shows the eCO₂ induced change is expected to turn negative? Maybe an average of when the 5 models show that happening – a broad description would be fine (mid-century, end of century, etc.).

Response and **action taken**: eCO₂-induced climate change impact on global vegetation carbon uptake has turned negative in the early 21st century, and is expected to be firmly negative thereafter. In the revised version, we have added this temporal information in the Abstract (see line 27).

2. Could you provide a table with model scenario descriptions and ensemble of models used in the main text? I see that the models used are in the SI – as a reader I would prefer that to be in the main text. It could be a truncated version of the detailed table, with the detailed table still being available in the SI.

Response and **action taken**: In the revised version, we have added the new Tables 1 and 2 into the main text, to better elaborate the CMIP6 factorial model simulations and ensemble of CMIP6 ESMs used in this study.

Table 1 | Information of CMIP6 ESMs used in this study.

Model Name	Land Surface Component	Modelling Center	Soil depth (m)
ACCESS-ESM1-5	CABLE2.4 with CASA-CNP	Commonwealth Scientific and Industrial Research Organisation, Australia	2.872
CanESM5	CLASS-CTEM	Canadian Centre for Climate Modelling and Analysis	4.1
CNRM-ESM2-1	ISBA-CTRIP	Centre National de Recherches Meteorologiques, France	10
E3SM-1-1	ELM1.1	U.S. Department of Energy	35.18
MIROC-ES2L	MATSIRO with VISIT-e	Japan Agency for Marine-Earth Science and Technology	14
MRI-ESM2-0	HAL1.0	Meteorological Research Institute of the Japan Meteorological Agency	8.5
UKESM1-0-LL	JULES-ES-1.0	U.K. Natural Environment Research Council, and Met Office	2

Table 2 | Description of CMIP6 factorial simulations.

Simulation name	Type	Forcing constraints		
		CO ₂ radiative forcing	CO ₂ physiological forcing	Other forcings
historical (1850-2014)	Fully-coupled mode	Yes, CO ₂ increases from 285 ppm to 397 ppm	Yes, CO ₂ increases from 285 ppm to 397 ppm	Yes, factors including CH ₄ , N ₂ O, aerosols, and land use vary over time
hist-bgc (1850-2014)	Biogeochemically-coupled mode	No, CO ₂ fixed at 285 ppm (pre-industrial level)	Yes, CO ₂ increases from 285 ppm to 397 ppm	Yes, factors including CH ₄ , N ₂ O, aerosols, and land use vary over time
hist-CO ₂ (1850-2014)	Single-forcing mode	Yes, CO ₂ increases from 285 ppm to 397 ppm	Yes, CO ₂ increases from 285 ppm to 397 ppm	No, factors except CO ₂ fixed at the pre-industrial level
ssp585 (2015-2100)	Fully-coupled mode	Yes, CO ₂ increases from 397 ppm to 1135 ppm	Yes, CO ₂ increases from 397 ppm to 1135 ppm	Yes, factors including CH ₄ , N ₂ O, aerosols, and land use vary over time
ssp585-bgc (2015-2100)	Biogeochemically-coupled mode	No, CO ₂ fixed at 285 ppm (pre-industrial level)	Yes, CO ₂ increases from 397 ppm to 1135 ppm	Yes, factors including CH ₄ , N ₂ O, aerosols, and land use vary over time

3. The authors did a nice job addressing the limitations of ESMs not representing disturbances well, and I appreciate the paragraph explaining how this may be a limitation of their results (lines 325-344). The authors explain how observation-based estimates are lower than model simulations in the historical period. Could the authors add a sentence here hypothesizing how that may impact their future simulation results (i.e., do they expect their results would show a greater decline if disturbances and biotic agents were better represented in models?)

Response and **action taken**: According to the reviewer's comment, we have elaborated on this issue in the Discussion section of the revised version of the manuscript. Details are as follows: "Considering that disturbance regimes are expected to intensify in many parts of the globe because of climate change (Seidl et al., 2017; McDowell et al., 2020; Senf and Seidl, 2021), the enhanced representation of the phenomena in ESMs would plausibly produce an even stronger decline in the global projection of the indirect CO₂ effect." (see lines 341-344).

References:

1. McDowell, N. et al. Pervasive shifts in forest dynamics in a changing world. *Science* **368**, eaaz9463 (2020).
2. Seidl, R. et al. Forest disturbances under climate change. *Nat. Clim. Change* **7**, 395-402 (2017).
3. Senf, C. & Seidl, R. Mapping the forest disturbance regimes of Europe. *Nat. Sustain.* **4**, 63-70 (2021).

REVIEWERS' COMMENTS

Reviewer #1 (Remarks to the Author):

Thank you for updating the manuscript based on my comments. They were addressed extremely thoroughly. Nice work.